# Does In-Context Learning Really Learn? Rethinking How Large Language Models Respond and Solve Tasks via In-Context Learning

**Quanyu Long**[*1]  **Yin Wu**[*1]  **Wenya Wang**[1]  **Sinno Jialin Pan**[1,2]

[1]Nanyang Technological University, Singapore  [2]The Chinese University of Hong Kong
{quanyu001, wuyi0023, wangwy}@ntu.edu.sg    sinnopan@cuhk.edu.hk

## Abstract

In-context Learning (ICL) has emerged as a powerful capability alongside the development of scaled-up large language models (LLMs). By instructing LLMs using few-shot demonstrative examples, ICL enables them to perform a wide range of tasks without updating millions of parameters. However, the precise contributions of demonstrations towards improving end-task performance have not been thoroughly investigated in recent analytical studies. In this paper, we empirically decompose the overall performance of ICL into three dimensions, label space, format, and discrimination, and we evaluate four general-purpose LLMs across a diverse range of tasks. Counter-intuitively, we find that the demonstrations have a marginal impact on provoking discriminative knowledge of language models. However, ICL exhibits significant efficacy in regulating the label space and format, which helps LLMs respond to desired label words. We then demonstrate that this ability functions similar to detailed instructions for LLMs to follow. We additionally provide an in-depth analysis of the mechanism of retrieval helping with ICL. Our findings demonstrate that retrieving the semantically similar examples notably boosts the model's discriminative capability. However, we also observe a trade-off in selecting good in-context examples regarding label diversity[1].

## 1 Introduction

Recent advancements in Large Language Models (LLMs) through in-context learning (ICL) have shown significant capability across a broad range of tasks (Brown et al., 2020; Min et al., 2022; Yoo et al., 2022; Pan et al., 2023; Wang et al., 2023a). By leveraging a few demonstrations (comprising input-label pairs), LLMs achieve high performance compared to zero-shot inference without updating millions of model parameters. Recent works have attempted to reveal the myth beneath ICL characteristics. Xie et al. (2022) propose that in-context demonstrations can enhance the model to "recall" the latent knowledge acquired during pre-training. Other empirical studies try to answer how ICL helps downstream tasks by studying the correctness of input-label mapping within the demonstrations (Min et al., 2022; Yoo et al., 2022; Pan et al., 2023). However, all those studies examine and report the gap between ICL and zero-shot setting, but do not provide a definitive answer regarding the specific factors that contribute to this gap. A more thorough exploration of the precise contributions of demonstrations towards improving end-task performance is necessary.

In this paper, we take a deeper look into the components of ICL's contribution and try to answer the following question: Which aspect of ICL power plays a crucial role? To achieve this, we decompose the improvement with ICL into three factors that empower the ICL ability of LLMs and quantitatively analyze their impact. These factors are *label space*, *label format*, and *discrimination*. The motivation stems from the tendency of general-purpose LLMs that produce responses with redundant information and inconsistent formats.

---

[*]Equal contribution.
[1]Codes are available at https://github.com/ruyue0001/decompose_ICL_improvement.

Observations from ICL applications indicate that incorporating ICL can help regulate LLM outputs to comply with designated label space and adhere to the format of demonstrative examples. Beyond the label space and format power of ICL, the discrimination power of ICL represents the model's discriminative ability to solve tasks provoked by semantically rich demonstrations. As ICL provides more few-shot contexts and examples, the LLMs are expected to "learn" from those contexts, thereby enhancing their discriminative capability and increasing the accuracy of predictions. However, it remains unclear whether the performance improvement brought by ICL is largely due to format/space regulation, or the capability of recalling latent discriminative knowledge via semantically rich demonstrations. In this paper, we quantify the contribution of each of the above-mentioned factors. The detailed definition for each factor and the quantification method are introduced in Section 3.

We experiment with four general-purpose and instruction-tuned LLMs and measure the three contributing factors on several classification, sequence labeling, and generation datasets. With extensive experiments, we aim to answer the following research questions: 1) Which aspect does ICL contribute to: discrimination, label space, or label format? 2) What is the mechanism of retrieval helping with ICL? 3) Beyond format, to what extent does demonstration text style affect generation tasks? Here are some key takeaways:

- A large part of the ICL improvement sources from the label space and format which are regulated by demonstrations. However, Counter-intuitively, ICL brings the least improvement on discrimination which also appears to be unstable across tasks.
- ICL functions similarly to detailed instruction in a prompt and serves the role of casting instruction of label space and format implicitly.
- When provided with random demonstrations, the knowledge of semantic discrimination is less invoked through those semantically rich contexts, which can even be harmful to confuse the model in many tasks and models.
- When provided with incorrect labels within the demonstrations, the ICL's power to regulate label space and format is barely influenced. This observation can explain the reason that incorrect labels within demonstrations have minimal impact on overall performance (Min et al., 2022).
- When retrieving the most similar examples as demonstrations, the discrimination power of ICL significantly improves. Our experiments show that LLM predictions align closely with the labels of retrieved demonstrations, with the majority class among these labels often matching the ground-truth label. However, when all the retrieved demonstrations are from the ground-truth class (lacks diversity), ICL's regulation power on label space and format will be weakened, suggesting that there is a trade-off when selecting good in-context examples.
- Similar to the observations in classification tasks that LLMs tend to follow the label space and format of the demonstrations, our findings in text generation tasks suggest the LLM responses also mimic the text style of demonstrations even when not explicitly instructed to do so.

## 2 Related Work

Recent studies on large language models (LLMs) have unveiled their capability for in-context learning (ICL), where the model adapts to new tasks solely through inference (Brown et al., 2020). Subsequent research studies have focused on both theoretical and empirical explorations to enrich the understanding of ICL's mechanisms. Xie et al. (2022) explain ICL as implicit Bayesian inference, where the pre-trained LMs implicitly infer and recover a latent concept that is learned during the pretraining. Similarly, Wang et al. (2023b) examine the ICL phenomenon through a Bayesian lens and view them as implicit topic models that infer a latent variable from prompts. Other theoretical studies investigate ICL based on learning algorithms for linear models on transformers, positing that ICL effectively operates as implicit gradient descent to update an "inner model" (Akyürek et al., 2023; Von Oswald et al., 2023; Dai et al., 2023). Recent work hypothesizes label words in demonstrations function as pivotal anchors that facilitate the aggregation and distributing of task-specific information (Wang et al., 2023a).

For empirical studies, Min et al. (2022) indicate that maintaining the structured format of demonstration (text-label pair) is critical, while random substitution of labels within demonstrations has minimal impact on performance. However, Yoo et al. (2022) challenge that ground-truth labels play a crucial role in ICL on the downstream tasks. Recent work evaluates the model's capability of task recognition through the introduction of wrong and abstract labels (Pan et al., 2023). From existing empirical studies, we can observe they primarily focus on the label correctness of demonstrations. However, the experiments using randomized labels within demonstrations fall short of elucidating the underlying ICL mechanism comprehensively. There is also a limited understanding regarding why incorrect labels have minimal impact on performance. Additionally, previous works examine and report the gap between the zero-shot setting and various ICL setups (e.g., ground-truth labels, random labels), while the factors underlying this gap and the mechanisms driving the efficacy of ICL remain ambiguous. We provide a thorough discussion of the relationships and distinctions in comparison to prior research in Appendix E, including differences in the definition of "Format" and the use of instruction-tuned models).

## 3  Decomposing ICL Improvement

### 3.1  True power of ICL may not be reflected in the observed performance gain

When querying general-purpose LLMs with specified downstream tasks, they may not strictly follow the instructions and are likely to generate responses with undesired formats. To evaluate the LLMs' performances under such circumstances, it is common to leverage post-processing scripts with the aim of filtering irrelevant fragments in the output and only keeping those relevant to the answer for identifying label verbalizers. However, simple post-processing may lead to inaccurate evaluations. Taking previous works (Min et al., 2022; Yoo et al., 2022) as an example, they test the presence of labels only at the first position of generated responses. However, instances with labels showing up in other positions would not be evaluated fairly. After performing ICL which provides text-label pairs (labels are single words) as demonstrations, larger amounts of predictions will be detected in the first position. This phenomenon is attributed to the tendency of LLMs to follow demonstration labels. Consequently, the true power of ICL is not properly evaluated.

To better quantify how ICL contributes to the performance gain and give precise attribution of the aforementioned tendency, we introduce two factors in ICL studies: label space and label format. **Label space** refers to the pre-defined set of label targets, encompassing all acceptable labels regardless of synonyms. **Label format** is the set of label verbalizers that could be identified by post-processing (exact string match), for example, NLI tasks consider a set of format patterns such as "non-entailment" and "not entail" within the post-process. It's worth mentioning that such post-processing scripts cannot cover all the format variations without prior knowledge, especially for those formats that occur less frequently. With the definition of label space and format, LLMs' outputs can be categorized into three types according to the post-processing, and Figure 1 gives an illustration:

- **OOS**: out-of-space, i.e., out of a pre-defined set of label targets. An example is predicting "neutral" in binary sentiment classification.
- **ISOOF**: in-space-out-of-format, i.e., out of the pre-defined format patterns of label verbalizers. For example, in NLI tasks, formats such as "no-entailment" or "none-entailment" which occur less frequently and are not included in the post-processing script will be considered as ISOOF.
- **ISIF**: in-space-in-format. Taking the above example, "non-entailment" or "not entail" can be categorized as ISIF. Note that only ISIF instances can contribute to correct predictions in the final evaluation.

In this study, we examine commonly encountered formats within our post-processing procedures, in accordance with the work by Qin et al. (2023). Examples of OOS and ISOOF for each task are listed in Appendix H. Through experiments, we observe that ICL has a strong ability to make the response follow the label space and format of the demonstrations. Such an effect of ICL can be summarized in Figure 1. Comparing the three types of outputs

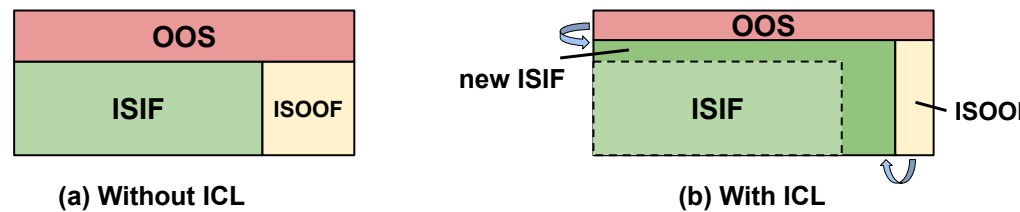

Figure 1: Inference instances can be categorized into three different sets, out-of-space (OOS), in-space-out-of-format (ISOOF) and in-space-in-format (ISIF). When performing ICL, a large proportion (almost all in our experiments) of OOS and ISOOF shift to ISIF.

with and without performing ICL, the proportion of ISIF outputs increases significantly by drawing samples from the original OOS and ISOOF categories. These **new ISIF** samples increase the amounts of right predictions, giving rise to higher performances. With this finding, we take the initiative to ask: how much of the overall performance gain brought by ICL is due to the power of regulating label space and label format?

### 3.2 Decomposing ICL improvements into label space, format, and discrimination

To answer the above question and give quantified attribution to label space and format, we first identify all the responses (w/o ICL and w/ ICL) as OOS, ISOOF, and ISIF. Then we track the change of categories for all instances (w/o ICL → w/ ICL). Figure 1 illustrates two main shift flows, namely OOS → ISIF and ISOOF → ISIF. In addition, we also observe a tiny portion of instances following inverse directions, ISIF → OOS and ISIF → ISOOF. Flows between OOS and ISOOF are not considered since they do not affect the observed ICL performance. By comparing the outputs w/o ICL and w/ ICL, we propose to decompose the overall performance enhancement facilitated by ICL into three contributing factors, label space, label format and discrimination, we define:

- **Label space power** as the performance gain brought by the power of ICL in regulating label space. It's calculated by $(n_{\mathrm{pred\_right}}^{\mathrm{OOS}\to\mathrm{ISIF}} - n_{\mathrm{pred\_right}}^{\mathrm{ISIF}\to\mathrm{OOS}})/N$, where $N$ is the total number of instances, and $n_{\mathrm{pred\_right}}$ represents the amount of correct predictions in ISIF. The calculation can be understood as the number of predictions corrected by ICL due to the change from OOS to ISIF minus those originally correct within ISIF but shifted to OOS afterward.

- **Label format power** as the performance gain brought by the power of ICL in regulating label format. Similar to label space, it is calculated by $(n_{\mathrm{pred\_right}}^{\mathrm{ISOOF}\to\mathrm{ISIF}} - n_{\mathrm{pred\_right}}^{\mathrm{ISIF}\to\mathrm{ISOOF}})/N$.

- **Discrimination power** as the performance gain brought by the change of predictions from wrong to right within the ISIF set, it is calculated by $(n_{\mathrm{W2R}}^{\mathrm{ISIF}} - n_{\mathrm{R2W}}^{\mathrm{ISIF}})/N$. W2R indicates those wrong predictions w/o ICL but are corrected w/ ICL. R2W refers to those correctly predicted w/o ICL but are misclassified to wrong labels w/ ICL. This measurement represents the model's discriminative ability provoked by semantically-rich demonstrations. As ICL provides more contexts and examples, the discriminative capability is expected to be enhanced.

## 4 Experiments Setup

### 4.1 Datasets

To be consistent with existing empirical studies (Min et al., 2022; Yoo et al., 2022; Pan et al., 2023), we evaluate the effectiveness of in-context learning on 9 classification datasets across 5 types of tasks, including: Sentiment Analysis: **SST-2** (Socher et al., 2013); Natural Language Inference: **WNLI** (Levesque et al., 2012) and **RTE** (Dagan et al., 2005; Haim et al., 2006; Giampiccolo et al., 2007; Bentivogli et al., 2009); Paraphrasing: Medical Question Pairs,

abbreviated as **MedQ** (McCreery et al., 2020) and **MRPC** (Dolan & Brockett, 2005); Hate Detection: **Tweet Hate** (Barbieri et al., 2020) and **Hate 18** (de Gibert et al., 2018), and Multi-class topic classification: **AG News** (Zhang et al., 2015) and **TREC** (Voorhees & Tice, 2000). In section 7, we experiment on four generation datasets: Story Generation: **ROCStories** and **ROCStories Ending** (Mostafazadeh et al., 2016), Text summarization: **Reddit** (Kim et al., 2019) and **SamSum** (Gliwa et al., 2019). We also experiment with several sequence labeling datasets, all the details and evaluation metrics are provided in Appendix A.

### 4.2 Models and other settings

We experiment with four general-purpose and instruction-tuned Large Language Models (LLMs): **ChatGPT** (OpenAI, 2024) (gpt-3.5-turbo-0613 version), **GPT-3** (Brown et al., 2020) (accessing via gpt-3.5-turbo-instruct), **Llama2** (Touvron et al., 2023) (llama2-13b-chat[2]) and **Mistral** (Jiang et al., 2023) (mistral-7b-instruct-v0.2[3]). We do not report the Llama2 scores on Hate Detection datasets due to the safety mechanism of Llama2. We use $k = 5$ demonstrations for all experiments in the paper, we analyze different numbers of $k$ in Appendix B. The demonstrations are selected from the training dataset of each task. If randomly selected, we experiment with 5 seeds and calculate averaged scores. Prompts and templates for each task can be found in Appendix H.

## 5 Which Aspect Does ICL Contribute to? Discrimination, Label Space or Label Format?

We investigate and take a deeper look at to what extent in-context learning (ICL) contributes to each specific aspect among discriminating power, label space, and label format. To answer this question, we first evaluate ICL using random demonstrations which are sampled randomly from the training dataset and denoted as:

**Random**: $k$ random demonstrations with ground truth labels, $\{(x_i, y_i)\}_{i=1}^k$, $(x_i, y_i) \in \mathcal{D}_{\text{train}}$. We present the results of classification tasks in this section, results for sequence labeling tasks are listed in Appendix D.

### 5.1 ICL is powerful to regulate the label space and format while disappointing regarding discriminating power

We measure the three contributing factors according to Section 3.2 on all classification tasks. Figure 2 illustrates the results of three different ICL powers using random demonstrations. From Figure 2 it is evident that the effects of label space and format are consistently positive across all models and tasks, and these two powers account for a large portion of the overall improvement brought by ICL, indicating ICL has strong power to regulate the label space and format, helping LLMs to respond and output desired label words and verbalizers. The label space demonstrates a trend that, as the number of classes increases, the contribution from label space occupies a larger portion. In multi-class classification tasks such as AG News and TREC, the label space has a greater impact than the sum of the other two factors.

However, discrimination is the most unstable component. Despite the demonstrations providing semantically rich contexts, counter-intuitively, those contexts have a marginal impact on provoking discriminative knowledge of language models to solve tasks, and the predictive ability of models is not significantly improved through ICL. From Figure 2 we can observe there is at least one model suffering from negative discrimination on all datasets. In NLI and Paraphrase tasks which accept two sentences as input, discrimination is apparent for ChatGPT and comparable with label space and label format. However, for other tasks, discriminating power has minimal positive contribution, and even becomes negative for all models in Hate Detection and multi-class classification datasets.

---

[2] https://huggingface.co/meta-llama/Llama-2-13b-chat-hf
[3] https://huggingface.co/mistralai/Mistral-7B-Instruct-v0.2. For fair comparison, we do not use the Mixture-of-Expert version ("Mixtral").

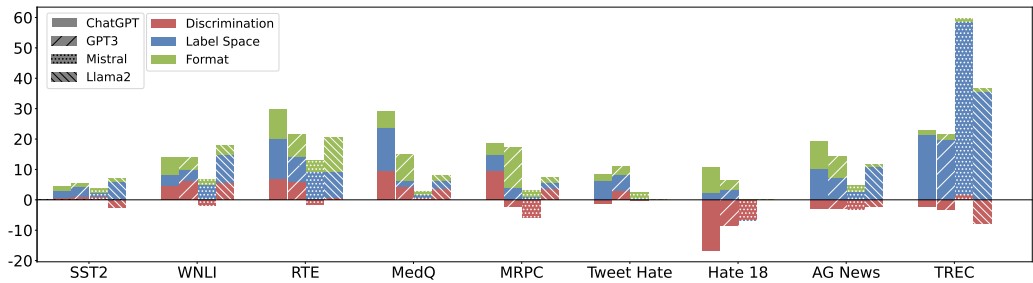

Figure 2: Classification results of decomposed ICL contribution: discrimination (red), label space (blue), label format (green) when using **Random** demonstrations. Scores below zero represent this factor has a negative effect on the performance. We find that discrimination power is the most unstable factor in ICL improvement.

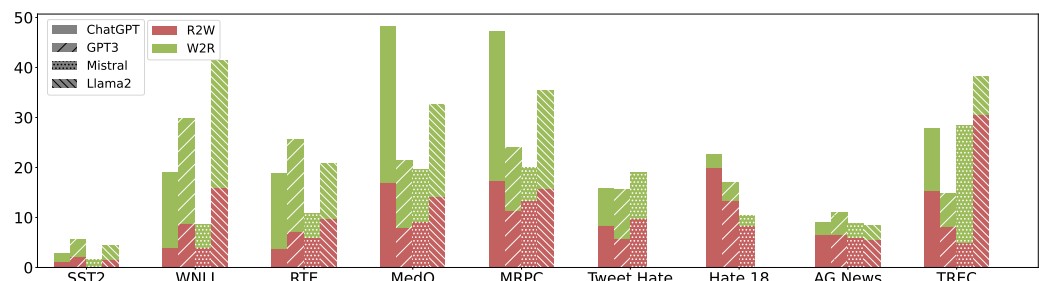

Figure 3: Right-to-Wrong (R2W) and Wrong-to-Right (W2R) percentage within the ISIF set. After performing ICL, R2W accounts for a large percentage surprisingly.

To explain why ICL brings the least improvement in discrimination power which also appears to be unstable across tasks, we compare the percentage of W2R (wrong-to-right) and R2W (right-to-wrong) instances within the ISIF set. These amounts are included in the calculation of discrimination power. Statistics are provided in Figure 3. With ICL, there are indeed a substantial amount of desired W2R cases, however, there is also a comparable proportion of undesired R2W cases for all of the 9 classification tasks. This result indicates that the impact of discriminating power oscillates when providing random demonstrations. ICL does not always make more correct predictions [4].

Previous works suggest that LLMs invoke "concepts" which are learned during pre-training through the demonstrations, and perform the implicit learning (Xie et al., 2022; Akyürek et al., 2023; Von Oswald et al., 2023). However, our results demonstrate that the knowledge of semantic discrimination is less invoked through the semantically-rich demonstrations, and these contexts can be even harmful to confuse the model to predict the correct label in many tasks and models. Nonetheless, this undesired effect can be mitigated by the powers of label space and format, which are dominant in the overall performance gain.

## 5.2 ICL Functions Similar to Detailed Instructions

With the above finding a large part of the ICL improvement sources from the label space and format (new ISIF), an intuitive question to ask is: when the amount of OOS and ISOOF is originally small for zero-shot setting, to what extent ICL could improve the performance? With the surge of instruction-tuned LLMs such as FLAN (Wei et al., 2022) and InstructGPT (Ouyang et al., 2022), we have witnessed the remarkable instruction-following capability of these LLMs. Given such capability, compared to ICL which regulates label space and format implicitly, they can be explicitly and directly incorporated into the task instruction which we refer to as *detailed instruction* (DI). For example, for NLI tasks, we add the following prompt to the instructions: *Please assign a label from ['entailment', 'non-*

---

[4]We conduct supplementary experiments to study why numerous examples in ISIF transition from right to wrong after performing few-shot ICL. The results and analysis are detailed in Appendix F.

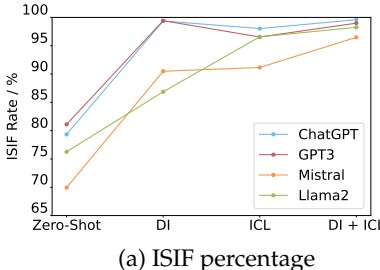 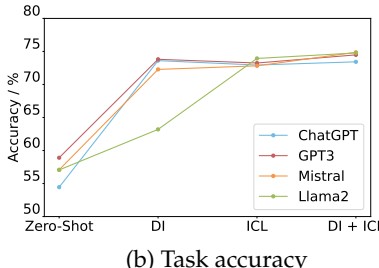

(a) ISIF percentage            (b) Task accuracy

Figure 4: Impact of DI (detailed instruction), ICL and their combination DI+ICL. Results are averaged scores across all classification tasks. Breakdown scores are provided in Appendix G. We observe that DI and ICL demonstrate similar performance and the benefit of ICL is nearly diminished when comparing the results of DI+ICL with ICL.

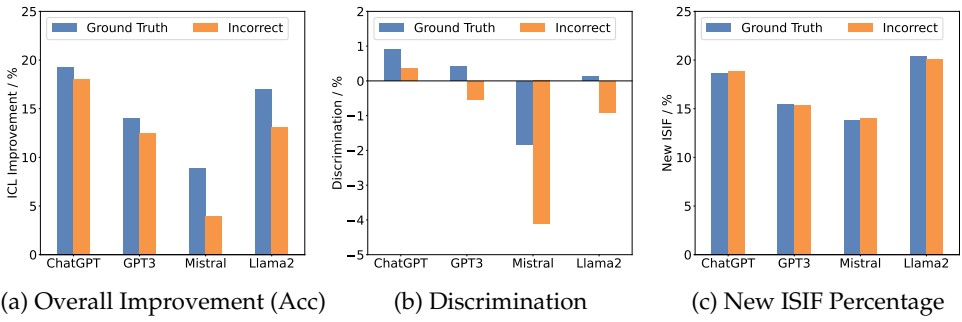

(a) Overall Improvement (Acc)     (b) Discrimination     (c) New ISIF Percentage

Figure 5: Impact of incorrect labels within the demonstrations compared to ground truth labels. Results are averaged scores across all classification tasks. (a) is the ICL overall improvement compared to the zero-shot setting; (b) is the decomposed discrimination score; (c) is the new ISIF percentage coming from OOS and ISOOF, this score can be viewed as the combination of label space and format. Figure (a) and (b) demonstrate a decrease in ICL performance and discrimination power when demonstrations contain incorrect labels, while the label space and format power remain unaffected in Figure (c).

*entailment'].* We then experiment two more prompt variations, DI and DI+ICL. Compared to the zero-shot setting (w/o ICL), DI adds detailed instructions to prompt (still zero-shot), ICL adds few-shot demonstrations which are randomly sampled, and DI+ICL is the setting using detailed instructions and ICL simultaneously.

Figure 4 illustrates the ISIF percentage and task accuracy of four settings (zero-shot, DI, ICL, and DI+ICL). As introduced in Section 3.1, a greater ISIF percentage indicates the label space and format of outputs adhere more closely to the pre-defined set, irrespective of label correctness. We can observe Figure 4 (a) and (b) have similar shapes, indicating the predominant impact of ICL power of label space and format. As shown in Figure 4, all three settings (DI, ICL, and DI+ICL) are having sufficient improvement in ISIF percentage and task performance. It is observed that DI and ICL exhibit comparable results in two figures, with Llama2 being an exception. This suggests that ICL functions similarly to detailed instructions and serves the role of casting instruction of label space and format implicitly. Additionally, by comparing DI with DI+ICL, the benefit of ICL is almost diminished, in contrast to the significant enhancements observed when comparing zero-shot and ICL. Particularly evident for ChatGPT and GPT3, the ISIF percentage in Figure 4 (a) even reaches 100% for DI alone. After providing demonstrations (DI+ICL), ICL's contribution to task accuracy is minimal as observed in Figure 4 (b). This suggests that when there exists greater space for improvement in OOS and ISOOF, the benefit of ICL becomes more apparent.

## 5.3 The Power of Space and Format is Consistent for Incorrect Labels

To substantiate our proposition that ICL brings limited discrimination power compared to label space and format, we conduct another set of experiments by replacing all the ground-

truth labels within demonstrations with incorrect labels (randomly selecting an incorrect label to replace). Prior work by Min et al. (2022) reveal that substitution of incorrect labels within demonstrations has minimal impact on task performance; however, the underlying reason for this phenomenon is not explained yet. In this section, we re-examine the influence of incorrect labels in ICL through an analysis of label space, format, and discrimination.

From Figure 5 (a), we observe the overall ICL improvement decreases across four models after changing demonstration labels. Figure 5 (b) and (c) elucidate the underlying mechanisms. Specifically, Figure (b) shows a pronounced decline in the ICL discrimination score, indicating incorrect label will substantially contaminate the discrimination power of ICL. Conversely, Figure (c) reveals that the proportion of new ISIF remains largely unaffected by incorrect labels, suggesting that the presence of such incorrect labels within demonstrations does not compromise the ICL's ability to regulate label space and format. This observation may account for the negligible impact on overall performance when substituting the incorrect labels. Since the powers of label space and format remain consistent under incorrect label settings, the discrimination power, despite being significantly affected, only occupies a small proportion compared to label space and format.

# 6 What is the Mechanism of Retrieval Helping with ICL?

It was established by prior work that when the whole training dataset is available, retrieving the demonstrations that are semantically similar to the input significantly enhances ICL (Liu et al., 2022). In our work, we take a deep look into how retrieval helps with ICL. For the retriever, we use SimCSE (Gao et al., 2021) to produce semantically meaningful sentence embeddings and cosine similarity to retrieve top-k (most similar) examples. We denote:
**Retrieval**: top-$k$ retrieved demonstrations, $\{(x_i, y_i)\}_{i=1}^{k}$ with highest $s(x', x_i)$, where $s(\cdot, \cdot)$ gives the similarity score, $x'$ is the current test input.

## 6.1 Retrieval helps discrimination ability of ICL

From the illustrated results in Figure 6 (a), we can observe retrieval improves the overall performance on all four models compared to randomly selecting demonstration. We then decompose the discrimination, label space, and format. Figure 6 (b) demonstrates the discrimination factor increases by a large margin (Retrieval v.s. Random). This suggests the predictive ability arises substantially through the retrieved semantically-similar examples. From Figure 6 (c), we do not observe an obvious difference in ISIF percentage between random demonstrations and retrieved demonstrations. Therefore, we conclude that retrieval mainly helps the discrimination ability, while brings limited enhancement on label space and format compared to randomly selecting demonstrations.

## 6.2 Why does retrieval help with discrimination? Is model prediction following the demonstrations' label?

When performing the demonstration retrieval, we observe a strong semantic correlation between the retrieved instances and the test input, often manifesting in label consistency. For instance, within the context of sentiment classification, given a positive input, the retrieved top demonstrations are all likely to have positive labels. Therefore, a natural question to ask is that: since almost all of the retrieved demonstrations have the same label, is model prediction following this majority label? To answer this question, suppose we cheat by acquiring the gold label $y'$ of the current input $x'$, we experiment with four additional methods of collecting demonstrations, which are:
**Homo-Random**: $k$ random demonstrations selected from the same class as $y\prime$.
**Homo-Retrieval**: top-$k$ retrieved demonstrations retrieved from the same class as $y\prime$.
**Hetero-Random**: $k$ random demonstrations selected from classes other than $y\prime$.
**Hetero-Retrieval**: top-$k$ retrieved demonstrations retrieved from classes other than $y\prime$.
Compared to the Homo and Hetero setting, the normal setting of **Random** and **Retrieval** would include demonstrations from all classes.

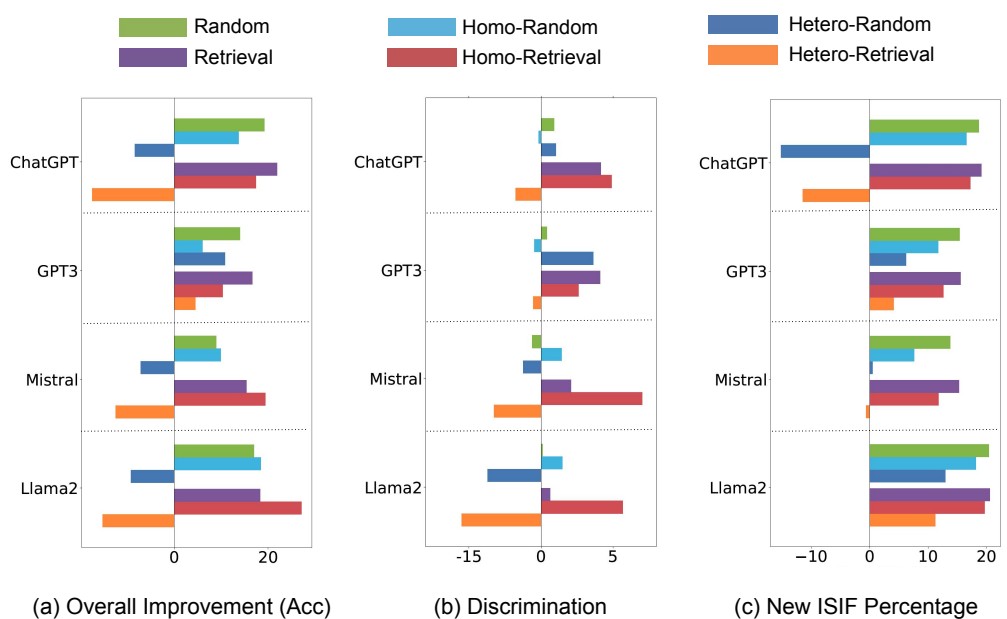

Figure 6: Comparing different methods of collecting demonstrations. Results are averaged scores across all classification tasks. Retrieval and Homo-Retrieval settings achieve the highest performance. In contrast, Hetero-Retrieval becomes detrimental, performing worse than Random selection. These findings suggest that retrieval mechanisms can fetch the most similar demonstrations which are likely to match the ground-truth label, and LLMs frequently generate responses that align with the labels of the retrieved demonstrations.

For Homo settings where demonstrations are drawn from the same categories as ground truth, Figure 6 (b) shows that ChatGPT and GPT-3 exhibit lower discrimination scores with Homo-Random demonstrations compared to Random ones. Conversely, Mistral and Llama2 demonstrate significantly higher scores. The disparity is especially pronounced in retrieval tasks, where Homo-Retrieval substantially outperforms standard Retrieval for Mistral and Llama2, enhancing discrimination capabilities notably when employing highly similar demonstrations sourced from identical categories. This finding suggests that, compared to Random selection within the ground-truth class, retrieval in Homo settings significantly augments the discrimination power of ICL, as the response of LLMs is more likely to follow this demonstrations' label (the ground-truth label in Homo settings) when performing retrieval. However, in Figure 6 (c) we can observe ISIF percentage decreases for Homo settings, indicating the powers of label space and format are weakened when all demonstrations have the same label. This finding underscores diversity also plays a critical role in selecting demonstrations, aligning with the research by Levy et al. (2023).

The results in Hetero settings can further support the aforementioned hypothesis. It is obvious that the Hetero settings significantly degrade performances. Firstly, we find that the discrimination power is notably compromised as shown in Figure 6 (b). When compared to random selection, retrieving from non-ground-truth classes proves more detrimental and exacerbates this decline. This suggests that LLM outputs tend to follow semantically similar demonstrations, which do not align with the ground truth in Hetero settings. Secondly, the absence of ground truth labels leads to uncertainty in determining the correct label token. This results in a decline in the new ISIF percentage (Figure 6 (c)). The above findings in Homo and Hetero experiments suggest that retrieval can fetch the most similar demonstrations which are likely to provide the correct label for LLM to follow and output.

## 7 Beyond Format, To What Extent Styles Affect Generation Tasks?

In previous sections, we discussed the regulation power brought by ICL on label space and format for classification tasks. To extend our investigation, we raise a question: could the

| Dataset | Zero-shot | ICL | Active | Formal | Passive |
|---|---|---|---|---|---|
| Reddit | 13.68 | **16.33** | 16.20 | 15.67 | 16.27 |
| SamSum | 25.50 | **28.81** | 27.96 | 26.74 | 27.98 |
| ROCStories | 4.92 | **7.73** | 7.68 | 6.93 | 7.35 |
| ROCStories Ending | 4.70 | **8.34** | 7.45 | 6.64 | 7.08 |

(a) ChatGPT

| Dataset | Zero-shot | ICL | Active | Formal | Passive |
|---|---|---|---|---|---|
| Reddit | 14.68 | **19.60** | 18.11 | 18.04 | 18.85 |
| SamSum | 26.40 | **32.29** | 30.12 | 27.57 | 29.83 |
| ROCStories | 4.46 | 7.89 | **8.61** | 7.85 | 7.29 |
| ROCStories Ending | 4.35 | **6.90** | 6.53 | 6.24 | 6.90 |

(b) GPT3

| Dataset | Zero-shot | ICL | Active | Formal | Passive |
|---|---|---|---|---|---|
| Reddit | 11.87 | 18.02 | **18.85** | 17.89 | 13.30 |
| SamSum | 22.68 | **30.12** | 29.79 | 27.53 | 28.01 |
| ROCStories | 6.28 | 7.97 | 8.24 | **8.27** | 7.72 |
| ROCStories Ending | 4.10 | **6.87** | 6.83 | 6.45 | 6.55 |

(c) Mistral

| Dataset | Zero-shot | ICL | Active | Formal | Passive |
|---|---|---|---|---|---|
| Reddit | 12.83 | **15.00** | 14.13 | 14.19 | 13.29 |
| SamSum | 24.97 | **28.89** | 28.76 | 27.97 | 28.38 |
| ROCStories | 7.74 | **9.23** | 9.20 | 8.23 | 8.88 |
| ROCStories Ending | 3.97 | **4.36** | 4.21 | 4.19 | 4.35 |

(d) Llama2

Table 1: Evaluation results for text generation datasets with different styles of references (labels) within the demonstrations. Active, Formal, and Passive denote the three editing styles. The table reveals that incorporating style shifts negatively impacts performance across all editing styles. These findings suggest the LLM responses also mimic the text style of demonstrations in generation tasks even when not instructed to do so.

regulation power on space and format affect generation tasks as well? In generation tasks such as story generation and text summarization, the ground-truth answers are inherently more flexible in their "formats" compared to classification tasks. In this context, we redefine the label format as the stylistic nuances of the generated text. We investigate whether the LLM output will follow the style and results in lower evaluation scores when altering the text styles of the demonstration labels. Since automatic evaluation methods for text generation such as BLEU score (Papineni et al., 2002) and ROUGE score (Lin, 2004) are based on n-grams, they are likely to be influenced by text styles.

A sentence could have multiple styles of expression while preserving its semantic meaning. We consider the following three style shifts: **Active**: transforming all references (labels) within the demonstrations to active voice; **Passive**: altering labels to passive voice; **Formal**: Adopting more formal vocabulary and grammar. We do not include the casual style since the references in employed summarization datasets are already casual. We randomly select 5 demonstrations from the training set and then prompt ChatGPT to modify the label styles of these demonstrations based on three specified directions. To ensure consistency to the original semantic meaning, we supplement this process with human effort. Detailed examples and case studies are provided in Appendix H for reference.

We present results from two text summarization and two story generation datasets. As shown in Table 1, the results indicate that ICL using original demonstration references achieves the highest scores across most cases. However, when incorporating style shifts, performance is hindered across all three settings, with the Formal setting experiencing the most pronounced decline. This outcome aligns with the intuition: vocabulary selection deviates more substantially from the ground truth when using the Formal setting, whereas the active and passive settings impose fewer changes in vocabulary. This observation underscores ICL's capacity to regulate response style (format) in generation tasks.

# 8   Conclusion

In this paper, we study the mechanisms underlying the effectiveness of ICL in improving end-task performance by decomposing the contributions of ICL into three factors: label space, label format, and discrimination. Our investigation reveals that ICL significantly improves performance by refining label space and format. Surprisingly, ICL yields the least improvement in eliciting discriminative knowledge within semantically-rich contexts. Additionally, our analysis of retrieving good demonstrations highlights the importance of choosing diverse and semantically relevant demonstrations to boost ICL performance. In summary, our study enhances comprehension regarding how LLMs respond and solve tasks via ICL and gives insights of selecting optimal demonstrations.

## 9 Acknowledgement

Sinno J. Pan thanks the support of the Hong Kong Jockey Club Charities Trust to the JC STEM Lab of Integration of Machine Learning and Symbolic Reasoning and the Microsoft Research Asia collaborative research grant. This research is also supported by the NTU Start-Up Grant (#023284-00001).

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

| Dataset | Train Size | Eval Size | Eval Split | Class Labels |
|---|---|---|---|---|
| SST2 | 67349 | 872 | Dev | positive, negative |
| RTE | 2490 | 277 | Dev | entailment, non-entailment |
| WNLI | 635 | 71 | Dev | entailment, non-entailment |
| MRPC | 3668 | 408 | Dev | equivalent, non-equivalent |
| Medical Question Pairs | 2438 | 610 | Random | equivalent, non-equivalent |
| Tweet Eval - Hate | 9000 | 1000 | Test | hate, non-hate |
| Hate Speech 18 | 9944 | 1000 | Random | hate, non-hate |
| AG News | 120000 | 1000 | Test (Sampled) | World, Sports, Business, Science & Technology |
| TREC | 5452 | 500 | Test | Abbreviation, Entity, Description and abstract concept, Human being, Location, Numeric value |

Table 2: Details of classification datasets. Training set is used for retrieval in Section 6. Evaluation is conducted using either the test or development split. In the absence of these splits, a random subset from the training set is sampled for evaluation.

| Dataset | Eval Size | Eval Split | Average Target Length (Words) |
|---|---|---|---|
| ROCStories | 500 | Test (Sampled) | 36.26 |
| ROCStories Ending | 500 | Test (Sampled) | 9.40 |
| Reddit | 563 | Test | 26.22 |
| SamSum | 819 | Test | 20.20 |

Table 3: Details of text generation datasets.

# A    More details about datasets and implementation

## A.1    Classification Datasets

Table 2 summarizes the classification datasets used in our study. We utilize 9 datasets across 5 tasks. Sentiment Analysis: **SST-2** (Socher et al., 2013); Natural Language Inference: **WNLI** (Levesque et al., 2012) and **RTE** (Dagan et al., 2005; Haim et al., 2006; Giampiccolo et al., 2007; Bentivogli et al., 2009); Paraphrasing: **MedQ** (McCreery et al., 2020) and **MRPC** (Dolan & Brockett, 2005); Hate Detection: **Tweet Hate** (Barbieri et al., 2020) and **Hate 18** (de Gibert et al., 2018), and Multi-class topic classification: **AG News** (Zhang et al., 2015) and **TREC** (Voorhees & Tice, 2000). All datasets in this paper are downloaded from Huggingface's Dataset (Lhoest et al., 2021).

For SST2, RTE, WNLI, MRPC, we utilize the GLUE Benchmark (Wang et al., 2019) version. Due to the absence of ground-truth labels in the GLUE test split, we rely on the validation split. Medical Question Pairs and Hate Speech 18 do not have official train-test splits. For Medical Question Pairs, we follow Min et al. (2022) and Yoo et al. (2022) to randomly split 20% (610 samples) as the evaluation set. For Hate Speech 18, we opt for a test set size of 1000 to ensure consistency with other datasets and manage API call costs. AG News has an official train-test split, but its test set size (7600) is significantly larger than those of other datasets. Thus, we randomly sample 1000 from the 7600. All classification datasets are evaluated using the Accuracy score.

| Dataset | Eval Size | Eval Split | Class Labels |
|---|---|---|---|
| SemEval 2014 Restaurants | 800 | Test | positive, negative, neutral, conflict |
| SemEval 2014 Laptops | 800 | Test | positive, negative, neutral, conflict |
| SemEval 2015 Restaurants | 685 | Test | positive, negative, neutral |
| SemEval 2016 Restaurants (English) | 676 | English-Test | positive, negative, neutral |
| CoNLL 2003 | 3684 | Test | person, location, organization, miscellaneous |
| WNUT 2017 | 1287 | Test | person, location, corporation, product, creative-work, group |

Table 4: Details of sequence labelling datasets. The label "conflict", denoting both positive and negative sentiments towards an aspect term, is uniquely annotated in SemEval 2014 and occurs infrequently (2.18% in restaurant reviews and 2.03% in laptop reviews).

## A.2 Text Generation Datasets

Table 3 outlines the statistics for the utilized text generation datasets. **ROCStories** and **ROCStories Ending** are derived from the same corpus (Mostafazadeh et al., 2016), each story precisely containing 5 sentences. In the ROCStories subset, the initial sentence serves as the prompt for generating the subsequent narrative, with the ground truth comprising the remaining 4 sentences. In contrast, the ROCStories Ending subset uses the first 4 sentences as input, with the model generating the final sentence. A test set of 500 stories is randomly selected for evaluation. The performance of the story generation datasets is assessed using the **BLEU-2** score (Papineni et al., 2002).

For text summarization datasets, **Reddit** (Kim et al., 2019) comprises informal documents from the online forum "Reddit". **SamSum** (Gliwa et al., 2019) consists of human-annotated dialogue summaries. Both datasets are evaluated using the **ROUGE-L** metric (Lin, 2004).

## A.3 Sequence Labelling Datasets

Table 4 provides an overview of the sequence labeling datasets utilized in our study. For ABSA, we use datasets from **SemEval 2014** Task 4 (Pontiki et al., 2014), **SemEval 2015** Task 12 (Pontiki et al., 2015) and **SemEval 2017** Task 5 (Pontiki et al., 2016). Following recent evaluations (Zhang et al., 2023), we select subsets including both restaurant and laptop reviews for SemEval 2014, restaurant reviews for SemEval 2015, and the English restaurant reviews subset for SemEval 2016. For NER, we employ the widely used **CoNLL 2003** (Tjong Kim Sang & De Meulder, 2003) and **WNUT 2017** (Derczynski et al., 2017) datasets. Evaluations of all sequence labeling datasets are conducted using the $F_1$ score based on exact matches of span-label pairs.

## A.4 Other Implementation Details

Inference on the Llama2 and Mistral models is conducted using PyTorch and Huggingface's transformers library. The `model.generate()` method with default parameters, including `temperature=1.0`, `top_k=50`, and `top_p=1.0`.

The ChatGPT generation is implemented through API call to `gpt-3.5-turbo` interface with the official `openai` python library. At the time of our experiments, `gpt-3.5-turbo` refers to the `gpt-3.5-turbo-0613` version, which is a snapshot dated June 13th 2023. The GPT3 generation is implemented by API call to `gpt-3.5-turbo-instruct` interface. According to OpenAI, this interface is a refined and instruction-tuned version of the old `text-davinci-003`

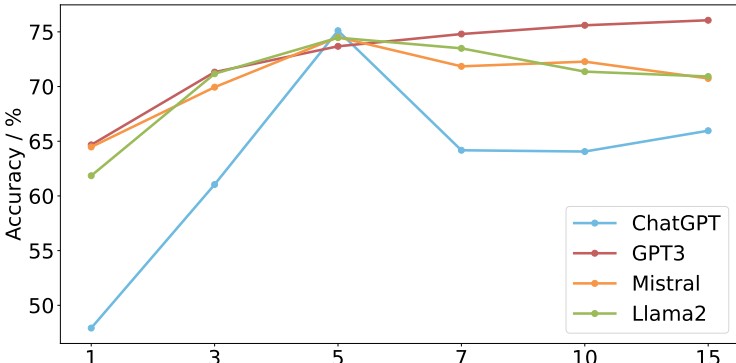

Figure 7: Average accuracy of all classification datasets with different number of demonstrations ($k$).

(GPT3) model. We maintain default decoding parameters of `temperature=1` and `top_p=1` for both.

## B  Number of demonstrations

We evaluate the performance of four LLMs across nine classification datasets for varying values of $k = 1, 3, 5, 7, 10, 15$. The mean accuracy across all datasets is illustrated in Figure 7. For ChatGPT, Mistral, and Llama2, optimal performance is observed at $k = 5$, yielding a convex accuracy curve. Conversely, GPT-3 reaches its highest accuracy at $k = 15$, exhibiting an upward trend with increasing $k$. These observations corroborate prior research (Liu et al., 2022). Consequently, for experimental consistency, we adopt $k = 5$ throughout our study.

## C  Measurements on sequence labeling tasks

We conducted experiments on sequence labeling tasks, specifically Named Entity Recognition (NER) and Aspect-Based Sentiment Analysis (ABSA). Detailed descriptions of the sequence labeling datasets utilized can be found in Appendix A.

Similar to classification tasks, we categorize the LLMs' outputs into three types according to the post-processing. Similarly, we use the notation IS / IF / OOS / OOF in Section 3.1, we denote:

- **OOF**: out-of-format. For NER and ABSA, the expected output format is a span-label pair. This consists of an entity span and its corresponding entity type for NER, and an aspect term span along with the sentiment toward the aspect term for ABSA. Responses that deviate into descriptive sentences with extraneous and redundant information, making post-processing challenging, are deemed out-of-format (OOF). For example, a response like "The sentence contains three entities 'Adam', 'Bob', 'Chris', these are all person names." is OOF. In contrast, responses such as "1. A. Parore - PERSON (Name) 2. C Ijaz Ahmad - PERSON (Surname or Last name)" and "Entities: Cuttitta, Italy. Type: Person, Organization" are considered In-Format (IF), since the span-label pairs are clearly identifiable. As discussed in Section 3.1, the post-processing script cannot accommodate all possible format variations.

- **IFOOS**: in-format-out-of-space. The outputs can be interpreted as span-label pairs, but the assigned class may not belong to the task's label space. For instance, predicting "Entity: Soccer — Type: Sports" for NER, or "bread, fantasitic" for ABSA. Here, "Sports" and "Fantasitic" are outside the defined label spaces for these tasks.

- **ISIF**: in-space-in-format. We include some synonyms in broad sense as IS. For instance, in zero-shot scenarios, models frequently classify location entities that are country names as "country" rather than "location". In this context, "country" is

considered as a synonym for "location" and is deemed ISIF if the output maintains the desired format.

As discussed in Section 3.1 and 3.2, the decomposition can be calculated in terms of the difference in IS/IF/OOS/OOF numbers w/ and w/o ICL. The granularity of prediction varies between classification and sequence labeling tasks. In classification tasks, each question has only one answer. For every question, predictions in zero-shot and ICL settings align, enabling us to track the change of label and shift of OOS, OOF and ISIF for each instance.

For sequence labeling tasks, each sentence may yield multiple predicted span-label pairs. Models often produce varying spans in zero-shot and ICL settings, leading to discrepancies in the number of predicted pairs. Therefore, the predicted pairs in zero-shot setting and ICL setting are not matched, making it challenging to track the shift of a predicted pair (e.g. OOS → ISIF). We can only indirectly measure the decomposed contribution through the reduction of IFOOS or IF-WrongSpan pair counts. For sequence labelling tasks, we also decompose the contribution of ICL into discrimination, label space and format:

- **Label Format**: the contributing factor from ICL in regulating response format, specifically to response in span-label pairs. It is simply calculated by $(n_{\text{zero}-\text{shot}}^{\text{OOF}} - n_{\text{ICL}}^{\text{OOF}})/S$, where $S$ being the total number of test set samples.

- **Label Space**: the contributing factor from ICL in regulating label space. It is calculated by $(n_{\text{zero}-\text{shot}}^{\text{IFOOS}} - n_{\text{ICL}}^{\text{IFOOS}})/S$.

- **Discrimination**: the contributing factor from ICL in correcting ISIF but wrong span / wrong class predictions. It is calculated by

$$((n_{\text{zero}-\text{shot}}^{\text{ISIF}_{\text{WrongSpan}}} - n_{\text{ICL}}^{\text{ISIF}_{\text{WrongSpan}}}) + (n_{\text{zero}-\text{shot}}^{\text{ISIF}_{\text{RightSpan}}^{\text{WrongLabel}}} - n_{\text{ICL}}^{\text{ISIF}_{\text{RightSpan}}^{\text{WrongLabel}}}))/S$$

  That is, the decrease in number of ISIF predictions but with wrong span, plus the decrease in number of ISIF predictions with right span but wrong label.

- **Indistinguishable**: It is important to note that the aforementioned three factors are derived from the reduction of False Positive (FP) predictions from various angles. Consequently, the count of True Positive (TP) predictions also increases. Although in certain cases, the increase in TP predictions can be directly attributed to the three factors (e.g., predicting right span wrong class in zero shot, corrected to right span right class with ICL), such cases are rare, with most instances remaining indistinguishable.

Given the task's characteristic where the class label is tied to the span, the aforementioned metrics estimate three decomposed factors. The label space factor is inevitably overlaps with the discrimination factor: insufficient label space information in zero-shot settings leads to wrong span (e.g., model is unable to know "Sports" is not within label space, hence mislabeling "Soccer" as an entity). Since the number of predicted pairs varies, we set the denominators to match the test set sizes when comparing with and without ICL to accurately reflect the dataset's relative percentages.

## D   Results on sequence labeling tasks

Figure 8 presents the decomposed results based on the method detailed in Appendix C. Surprisingly, in format-sensitive tasks like NER and ABSA, the ICL's role in format regulation is minimal compared to other factors. This may be because NER and ABSA task instructions inherently convey more format details than classification tasks (see Appendix H for examples). Here, the instructions are containing some hint for response format ("Please identify all named entities and classify their types"), but not containing any label space information. As discussed in Section 5.2, detailed instructions already offer sufficient format and label space information, making additional demonstrations marginally beneficial.

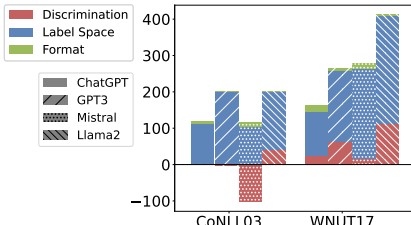 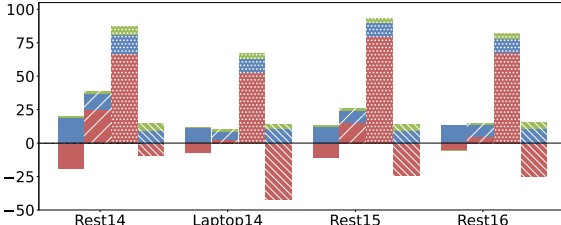

Figure 8: Decomposed ICL contributing factors scores for sequence labelling datasets. Rest14: SemEval 2014 - Restaurants subset; Laptop14: SemEval 2014 - Laptops subset; Rest15: SemEval 2015 - Restaurants subset; Rest16: SemEval 2016 - Restaurants subset. The resulting scores do not quantitatively correspond to improvements in $F_1$ scores; rather, they should be interpreted in terms of the relative proportions of the three contributing elements.

The proportion of label space varies between NER and ABSA datasets, especially in zero-shot setting, the label space for named entity types is considerably larger than that for sentiment types in ABSA dataset. This is evident in the WNUT17 dataset, where the label space differs substantially from that of CoNLL03. For instance, It has type "corporation" specifically for name of company / corporation, "creative-work" for name of artwork / music album, etc. Without ICL, it becomes exceedingly challenging for models to extract such entities and assign correct classes. Conversely, sentiment types in ABSA datasets are less ambiguous and more straightforwardly annotated.

Discrimination factors vary across models and datasets. ChatGPT consistently exhibits low or negative discrimination scores. Conversely, GPT-3 shows high discrimination scores. Mistral and Llama2 present differing behaviors: Mistral has a negative discrimination score for NER but positive for ABSA, while Llama2 shows the reverse pattern.

# E   Discussion on Relation and Difference to Previous Works

Min et al. (2022) aim to explore the influence of input-label mapping and the format of such mapping (e.g., only input, only label). Their experiments involve providing incorrect labels in demonstrations, we discuss this setting in our Section 5.3. However, despite using the same term "format", the definition and researching focus is different. Their experiments on formats are formulated as providing demonstration texts without labels and labels without text, i.e., the format of demonstrations (format of the input-label pairing pattern). Their analytical scope on "format" can be interpreted as how format of demonstrations affect performance, we instead study the label/response format and the regulation effect brought by ICL.

Pan et al. (2023) employ the terms "Task Recognition" (TR) and "Task Learning" (TL) as two factors influencing LLMs' few-shot capability. TR denotes the model's ability to perform effectively without depending on input-label pairings. The model can maintain good performance even with incorrect input-label mappings, this resembles previous work Min et al. (2022). TL can be conceptualized the "label space" power in our paper. Their experiments on TL are are limited to altering the label space (such as converting "positive"/"negative" labels to 0/1 or other symbols). However, since the models they adopted are not instruction-tuned, they wouldn't be able to explore the regulation effect on response format. This is one major difference between our work and these previous works, as the response generated by current general-purpose, human-instruction-aligned LLMs differ from the early models.

Both works point out the intriguing phenomenon that contaminating demonstration label correctness and replacing label words that have semantically-rich information will affect the model performance. However, their work lack detailed and quantitaive analysis on decomposing the contributions of factors to ICL. Our focus on ICL's label space and format regulation effect lies in studying the ability of changing OOS and OOF label to desired labels

within the pre-defined set, and we aim to separate such effect from the ability of assigning correct label, i.e. discrimination.

Regarding retrieving semantically-similar demonstrations, Lyu et al. (2023) find the responses of LLMs are more likely to follow the labels of demonstrations that are semantically close to the input and describe this phenomenon as "Copy Effect". Here we summarize the difference between our work and Lyu et al. (2023) as the following: (1) Difference of target aspects being studied. The "Copy Effect" is discussed under the context of using incorrect labels in demonstrations. As we mentioned in Section 2, all previous works focus on the label correctness, and take it as the start point to study the ICL. Instead, we decompose the benefit from ICL into three factors and take it as the start point to study which aspect ICL contributes to. (2) We take a deeper look into the "Copy Effect" from the perspective of majority label class instead of false labels. As stated in Section 6.2, we find that when providing demonstrations with the same class of current query's ground truth answer ("homo" setting), we can observe ISIF percentage decreases, indicating the powers of label space and format are weakened when all demonstrations have the same label (especially semantically close to the input). This finding underscores diversity also plays critical role in selecting demonstrations.

# F  Why do models change from correct to incorrect predictions after performing few-shot ICL?

As we discussed in Section 5.1, the benefit from ICL regarding discriminating power is unstable. Notably, our observations revealed a phenomenon rarely addressed in prior ICL research: when comparing predictions in zero-shot and ICL settings, there are comparable proportion of cases that changes correct predictions to wrong answers.

Our hypothesis posits that the quality of demonstrations significantly impacts ICL performance, and random examples may be unrelated or even detrimental to the prediction of some instances. We collect the statistics on another set of experiments in retrieval setting, where ICL demonstrations are selected based on the retriever (Section 6.1). Results indicate that the R2W rate can be moderately mitigated by retrieved demonstrations, as evidenced in Table 5. We observe an improvement in the differences between W2R and R2W rate for all models utilizing retrieved demonstrations. However, the R2W rate remains significant even with retrieved semantically-similar demonstrations. Potential explanations include: 1) the retrieval method based on semantic similarity is imperfect; 2) demonstrations can be regarded as "additional parameters" that influence the token generation probability distribution. We plan to explore this aspect in future work.

| Category | Random | | | | Retrieved | | | |
|---|---|---|---|---|---|---|---|---|
| | ChatGPT | GPT3 | Mistral | Llama2 | ChatGPT | GPT3 | Mistral | Llama2 |
| R2R | 60.40% | 65.67% | 71.23% | 63.77% | 61.10% | 65.01% | 71.80% | 65.13% |
| W2W | 16.10% | 15.99% | 14.64% | 10.29% | 13.15% | 12.72% | 10.96% | 10.24% |
| W2R | 13.13% | 10.34% | 7.25% | 12.47% | 15.66% | 13.72% | 10.97% | 12.47% |
| R2W | 10.37% | 7.96% | 6.88% | 13.40% | 10.09% | 8.55% | 6.26% | 12.16% |
| W2R-R2W | 2.76% | 2.38% | 0.37% | -0.93% | 5.57% | 5.18% | 4.71% | 0.32% |

Table 5: Comparison of R2R, W2W, W2R and R2W ratio under Random and Retrieval setting, together with the difference in W2R and R2W. Results are averaged scores across 9 classification datasets.

# G    Breakdown scores of classification tasks

## G.1    Results in Section 5.1

Table 6 and 7 offers detailed scores corresponding to the analysis in Section 5.1. Table 6 details the scores for discrimination, label space, and format across four models and nine classification datasets, including an additional column for the average scores across these datasets.

| Model | Factor | SST2 | WNLI | RTE | MedQ | MRPC | Tweet Hate | Hate 18 | AG News | TREC | AVG |
|-------|--------|------|------|-----|------|------|------------|---------|---------|------|-----|
| ChatGPT | Discrimination | 0.67% | 4.79% | 6.86% | 9.51% | 9.51% | -1.28% | -16.58% | -2.98% | -2.32% | 0.91% |
| | Label Space | 2.16% | 3.38% | 13.29% | 14.23% | 5.44% | 6.36% | 2.54% | 10.14% | 21.28% | 8.76% |
| | Format | 1.56% | 5.63% | 9.68% | 5.44% | 3.68% | 2.28% | 8.24% | 9.16% | 1.64% | 5.26% |
| GPT3 | Discrimination | 1.08% | 6.20% | 6.06% | 4.23% | -2.16% | 3.08% | -8.60% | -2.98% | -3.12% | 0.42% |
| | Label Space | 3.19% | 3.66% | 8.07% | 2.03% | 3.97% | 5.12% | 3.50% | 7.20% | 19.76% | 6.28% |
| | Format | 1.26% | 4.23% | 7.22% | 8.72% | 13.09% | 2.90% | 2.84% | 7.16% | 1.88% | 5.48% |
| Mistral | Discrimination | 1.06% | -1.97% | -1.52% | 0.62% | -5.98% | -0.36% | -6.84% | -3.20% | 1.80% | -1.82% |
| | Label Space | 1.24% | 5.07% | 9.46% | 1.28% | 1.03% | 0.28% | -0.06% | 2.70% | 57.00% | 8.67% |
| | Format | 1.54% | 1.69% | 3.54% | 0.72% | 2.11% | 2.02% | 0.10% | 2.14% | 0.80% | 1.63% |
| Llama2 | Discrimination | -2.68% | 5.63% | 0.65% | 3.61% | 3.58% | - | - | -2.22% | -7.76% | 0.12% |
| | Label Space | 6.17% | 9.14% | 8.81% | 2.85% | 2.21% | - | - | 10.94% | 35.52% | 10.81% |
| | Format | 0.89% | 3.10% | 11.12% | 1.48% | 1.47% | - | - | 0.56% | 1.00% | 2.80% |

Table 6: Decomposed ICL contribution factors for classification datasets. Results for hate speech detection using Llama2 are excluded due to its safety mechanisms hindering the generation of meaningful responses. This applies to the subsequent tables as well.

The analysis of the instability in discrimination power contribution is detailed in Table 7, with corresponding breakdown scores. Figure 3 highlights the proportions of R2W (right-to-wrong) and W2R (wrong-to-right). For comprehensive understanding, we included all four answer shift directions: R2W, W2R, R2R (right-to-right), and W2W (wrong-to-wrong).

| Model | Category | SST2 | WNLI | RTE | MedQ | MRPC | Tweet Hate | Hate 18 | AG News | TREC | AVG |
|-------|----------|------|------|-----|------|------|------------|---------|---------|------|-----|
| ChatGPT | W2R | 1.79% | 15.09% | 15.01% | 31.24% | 29.83% | 7.47% | 2.77% | 2.44% | 12.51% | 13.13% |
| | R2W | 1.04% | 3.86% | 3.88% | 17.03% | 17.39% | 8.43% | 19.95% | 6.52% | 15.25% | 10.37% |
| | R2R | 94.20% | 45.26% | 63.69% | 38.50% | 32.89% | 57.34% | 64.19% | 82.90% | 64.65% | 60.40% |
| | W2W | 2.97% | 35.79% | 17.42% | 13.22% | 19.89% | 26.76% | 13.09% | 8.13% | 7.59% | 16.10% |
| GPT3 | W2R | 3.45% | 20.92% | 18.28% | 13.41% | 12.62% | 9.80% | 3.69% | 4.40% | 6.52% | 10.34% |
| | R2W | 2.12% | 8.82% | 7.29% | 8.02% | 11.42% | 5.70% | 13.32% | 6.69% | 8.27% | 7.96% |
| | R2R | 91.50% | 34.97% | 62.05% | 66.88% | 61.46% | 52.66% | 61.48% | 82.21% | 77.79% | 65.67% |
| | W2W | 2.94% | 35.29% | 12.37% | 11.30% | 14.50% | 31.84% | 21.51% | 6.69% | 7.43% | 15.99% |
| Mistral | W2R | 1.43% | 4.67% | 4.98% | 10.31% | 6.76% | 9.15% | 1.98% | 2.69% | 23.29% | 7.25% |
| | R2W | 0.19% | 4.00% | 5.90% | 9.22% | 13.31% | 9.86% | 8.37% | 6.02% | 5.02% | 6.88% |
| | R2R | 96.56% | 47.00% | 71.77% | 73.05% | 66.97% | 58.25% | 82.08% | 83.31% | 62.10% | 71.23% |
| | W2W | 1.82% | 44.33% | 17.34% | 7.41% | 12.96% | 22.74% | 7.57% | 7.98% | 9.59% | 14.64% |
| Llama2 | W2R | 2.66% | 25.29% | 10.89% | 18.42% | 19.53% | - | - | 2.80% | 7.72% | 9.70% |
| | R2W | 1.65% | 16.09% | 9.82% | 14.26% | 15.84% | - | - | 5.61% | 30.52% | 10.42% |
| | R2R | 92.48% | 42.91% | 67.88% | 58.20% | 55.65% | - | - | 77.99% | 51.27% | 49.60% |
| | W2W | 3.22% | 15.71% | 11.42% | 9.12% | 8.48% | - | - | 13.60% | 10.49% | 8.00% |

Table 7: Within ISIF set, percentages of samples being corrected (W2R), mislabelled (R2W), no change (R2R and W2W) with ICL demonstrations provided comparing to zero-shot.

## G.2 Results in Section 5.2

Table 8 and Table 9 correspond to the analysis in Section 5.2. The results are the breakdown scores of ISIF percentage and task accuracy for 4 models across 9 tasks, using different prompt variations: Zero-shot, detailed instruction (DI) , 5-shot in-context learning (ICL), and a combination of DI and ICL (DI+ICL). The demonstrations of ICL and DI+ICL are randomly sampled from the training dataset, and the results are averaged scores among those random seeds in these two few-shot settings.

The comparative analysis reveals that DI and ICL produce similar results in terms of ISIF percentage and task accuracy, with Llama2 as an exception. This indicates that ICL generally functions comparably to detailed instructions, implicitly providing guidance on label space and format. However, this is not universally true. For instance, ChatGPT with ICL significantly outperforms DI in ISIF percentage on the TREC dataset, whereas Llama2 with ICL performs worse than DI.

| Model | Prompt | SST2 | WNLI | RTE | MedQ | MRPC | Tweet Hate | Hate 18 | AG News | TREC | AVG |
|---|---|---|---|---|---|---|---|---|---|---|---|
| ChatGPT | Zero-shot | 95.07% | 80.28% | 69.31% | 75.09% | 88.24% | 89.40% | 85.30% | 70.40% | 60.20% | 79.34% |
| | DI | 99.81% | 100.00% | 99.88% | 100.00% | 100.00% | 100.00% | 100.00% | 97.73% | 96.87% | 99.37% |
| | ICL | 99.50% | 100.00% | 99.28% | 99.98% | 100.00% | 99.94% | 99.94% | 93.96% | 90.20% | 98.03% |
| | DI+ICL | 99.85% | 100.00% | 100.00% | 100.00% | 100.00% | 100.00% | 100.00% | 99.70% | 96.87% | 99.60% |
| GPT3 | Zero-shot | 91.86% | 87.32% | 79.42% | 83.93% | 78.43% | 86.70% | 87.20% | 74.10% | 61.00% | 81.11% |
| | DI | 98.85% | 100.00% | 100.00% | 99.51% | 99.51% | 98.40% | 99.90% | 99.40% | 99.20% | 99.42% |
| | ICL | 96.58% | 99.15% | 98.34% | 99.51% | 99.56% | 97.86% | 98.08% | 91.16% | 88.60% | 96.54% |
| | DI+ICL | 98.56% | 99.34% | 99.13% | 99.84% | 99.75% | 96.90% | 98.90% | 98.87% | 99.60% | 98.99% |
| Mistral | Zero-shot | 78.21% | 85.92% | 80.51% | 96.72% | 97.55% | 95.80% | 98.90% | 85.80% | 9.40% | 80.98% |
| | DI | 79.82% | 78.87% | 87.14% | 99.67% | 97.55% | 99.40% | 99.50% | 97.80% | 88.29% | 92.01% |
| | ICL | 80.87% | 97.18% | 97.11% | 99.25% | 99.66% | 99.64% | 99.24% | 92.12% | 86.04% | 94.57% |
| | DI+ICL | 85.97% | 100.00% | 97.86% | 99.84% | 100.00% | 99.80% | 99.70% | 99.04% | 99.74% | 97.99% |
| Llama2 | Zero-shot | 87.84% | 73.24% | 70.40% | 93.61% | 95.59% | - | - | 78.80% | 34.40% | 76.27% |
| | DI | 98.05% | 90.14% | 91.70% | 92.62% | 89.71% | - | - | 80.30% | 65.60% | 86.87% |
| | ICL | 95.73% | 96.62% | 96.46% | 99.05% | 99.85% | - | - | 93.54% | 94.76% | 96.57% |
| | DI+ICL | 98.92% | 98.03% | 95.74% | 98.52% | 98.28% | - | - | 99.28% | 98.96% | 98.25% |

Table 8: Breakdown scores of ISIF percentage for DI (detailed instruction), ICL and their combination DI+ICL.

| Model | Prompt | SST2 | WNLI | RTE | MedQ | MRPC | Tweet Hate | Hate 18 | AG News | TREC | AVG |
|---|---|---|---|---|---|---|---|---|---|---|---|
| ChatGPT | Zero-shot | 90.48% | 39.44% | 46.57% | 37.21% | 35.54% | 58.80% | 71.90% | 62.90% | 47.20% | 54.45% |
| | DI | 94.38% | 58.43% | 80.99% | 79.90% | 72.57% | 66.87% | 72.35% | 74.53% | 62.47% | 73.61% |
| | ICL | 94.91% | 57.46% | 77.18% | 79.28% | 71.86% | 66.58% | 68.06% | 72.59% | 68.56% | 72.94% |
| | DI+ICL | 95.49% | 59.38% | 76.41% | 79.31% | 72.31% | 65.10% | 68.46% | 74.23% | 70.20% | 73.43% |
| GPT3 | Zero-shot | 85.55% | 38.03% | 54.29% | 63.11% | 57.11% | 50.60% | 65.20% | 65.30% | 50.99% | 58.91% |
| | DI | 92.78% | 52.11% | 77.50% | 79.84% | 76.72% | 60.80% | 67.70% | 84.00% | 72.82% | 73.81% |
| | ICL | 91.19% | 60.56% | 78.77% | 78.39% | 74.84% | 62.10% | 63.30% | 78.02% | 72.04% | 73.25% |
| | DI+ICL | 92.20% | 62.81% | 79.42% | 78.43% | 69.85% | 61.19% | 66.40% | 85.10% | 74.92% | 74.48% |
| Mistral | Zero-shot | 75.23% | 43.66% | 62.45% | 79.34% | 78.19% | 65.30% | 89.46% | 76.40% | 6.00% | 64.00% |
| | DI | 76.61% | 46.48% | 74.64% | 80.82% | 77.94% | 63.30% | 90.16% | 79.60% | 67.26% | 72.98% |
| | ICL | 78.90% | 51.27% | 74.73% | 83.05% | 77.84% | 66.88% | 83.23% | 78.50% | 65.08% | 73.28% |
| | DI+ICL | 80.64% | 55.56% | 76.79% | 87.21% | 79.41% | 67.00% | 88.05% | 79.30% | 70.65% | 76.07% |
| Llama2 | Zero-shot | 82.45% | 45.07% | 54.87% | 63.28% | 60.05% | - | - | 65.70% | 28.00% | 57.06% |
| | DI | 87.16% | 50.70% | 72.20% | 60.16% | 58.09% | - | - | 62.70% | 51.40% | 63.20% |
| | ICL | 89.43% | 66.76% | 75.45% | 78.16% | 75.78% | - | - | 75.02% | 56.96% | 73.94% |
| | DI+ICL | 87.20% | 62.82% | 76.39% | 72.30% | 76.47% | - | - | 73.84% | 74.36% | 74.77% |

Table 9: Breakdown scores of task accuracy for DI (detailed instruction), ICL and their combination DI+ICL.

### G.3 Results in Section 5.3

Table 10 presents the analysis in Section 5.3, comparing the scores of ICL accuracy improvements, discrimination scores and new ISIF percentages. GT represents that the demonstrations are randomly sampled with ground-truth labels, consistent with Section 5.1. To investigate the minimal impact of incorrect label substitutions within demonstrations on task performance (Min et al., 2022), we conduct another set of experiments by replacing all the ground-truth labels within demonstrations with incorrect labels (randomly chosen). As shown in the Table 10, the New ISIF percentage remains largely unaffected.

| Dataset | ChatGPT | | GPT3 | | Mistral | | Llama2 | |
|---|---|---|---|---|---|---|---|---|
| | GT | Incorrect | GT | Incorrect | GT | Incorrect | GT | Incorrect |
| SST2 | 4.43% | 3.83% | 5.64% | -5.00% | 3.67% | -6.03% | 7.87% | 5.92% |
| WNLI | 18.03% | 21.97% | 22.54% | 25.92% | 7.61% | 3.66% | 21.69% | 19.44% |
| RTE | 30.61% | 29.39% | 23.90% | 23.83% | 12.27% | 8.38% | 20.58% | 17.83% |
| MedQ | 42.07% | 41.54% | 15.28% | 16.66% | 3.70% | -0.20% | 14.89% | 11.90% |
| MRPC | 36.32% | 37.40% | 15.74% | 15.69% | -3.63% | -9.41% | 15.74% | 15.74% |
| Tweet Hate | 7.78% | 5.20% | 11.50% | 9.14% | 1.58% | -2.54% | - | - |
| Hate 18 | -3.84% | -4.02% | -1.90% | 3.08% | -6.22% | -4.30% | - | - |
| AG News | 16.46% | 15.20% | 12.72% | 11.62% | 2.10% | -0.90% | 9.32% | 7.16% |
| TREC | 21.36% | 12.12% | 20.64% | 11.04% | 59.08% | 46.56% | 28.96% | 13.60% |
| AVG | 19.25% | 18.07% | 14.01% | 12.44% | 8.91% | 3.91% | 13.23% | 10.18% |

(a) Overall Improvement (Acc) compared to zero-shot

| Dataset | ChatGPT | | GPT3 | | Mistral | | Llama2 | |
|---|---|---|---|---|---|---|---|---|
| | GT | Incorrect | GT | Incorrect | GT | Incorrect | GT | Incorrect |
| SST2 | 0.67% | -0.21% | 1.08% | -9.45% | 1.06% | -2.06% | -2.68% | -4.91% |
| WNLI | 4.79% | 6.48% | 6.20% | 7.32% | -1.97% | -5.92% | 5.63% | 7.61% |
| RTE | 6.86% | 6.50% | 6.06% | 6.14% | -1.52% | -4.04% | 0.65% | -1.66% |
| MedQ | 9.51% | 9.28% | 4.23% | 5.70% | 0.62% | -2.43% | 3.61% | 0.59% |
| MRPC | 9.51% | 11.23% | -2.16% | -2.45% | -5.98% | -12.30% | 3.58% | 3.82% |
| Tweet Hate | -1.28% | -3.60% | 3.08% | 0.62% | -0.36% | -2.76% | - | - |
| Hate18 | -16.58% | -15.62% | -8.60% | -3.76% | -6.84% | -4.30% | - | - |
| AG News | -2.98% | -3.68% | -2.98% | -4.02% | -3.20% | -3.56% | -2.22% | -3.14% |
| TREC | -2.32% | -7.04% | -3.12% | -4.96% | 1.80% | 0.28% | -7.76% | -10.44% |
| AVG | 0.91% | 0.37% | 0.42% | -0.54% | -1.82% | -4.12% | 0.12% | -1.16% |

(b) Discrimination

| Dataset | ChatGPT | | GPT3 | | Mistral | | Llama2 | |
|---|---|---|---|---|---|---|---|---|
| | GT | Incorrect | GT | Incorrect | GT | Incorrect | GT | Incorrect |
| SST2 | 4.43% | 4.66% | 4.72% | 5.71% | 2.66% | -4.72% | 8.65% | 8.74% |
| WNLI | 19.72% | 19.44% | 11.83% | 9.01% | 11.27% | 6.20% | 23.38% | 23.66% |
| RTE | 29.96% | 30.69% | 18.92% | 19.06% | 16.61% | 14.58% | 26.06% | 25.70% |
| MedQ | 24.07% | 24.07% | 15.57% | 15.25% | 2.52% | 2.56% | 5.44% | 5.28% |
| MRPC | 11.76% | 11.76% | 21.13% | 20.83% | 4.07% | 4.31% | 4.26% | 4.26% |
| Tweet Hate | 10.54% | 10.48% | 11.16% | 11.76% | 3.84% | 1.68% | - | - |
| Hate 18 | 14.08% | 9.02% | 10.88% | 7.62% | 0.34% | 0.34% | - | - |
| AG News | 23.56% | 23.46% | 17.06% | 18.28% | 6.32% | 3.34% | 14.74% | 14.72% |
| TREC | 30.00% | 30.84% | 27.60% | 20.56% | 76.64% | 72.80% | 60.36% | 58.32% |
| AVG | 18.68% | 18.27% | 15.43% | 14.23% | 13.81% | 11.23% | 20.41% | 20.10% |

(c) New ISIF percentage (label space + label format)

Table 10: Impact of altering demonstrations with incorrect labels. New ISIF percentage remains largely unaffected.

### G.4 Results in Section 6

Tables 11, 12 and 13 contain breakdown results as discussed in Section 6. Table 11 lists the overall accuracy improvements with ICL compared to the zero-shot setting, Table 12 provides the discrimination scores, and Table 13 presents the new ISIF percentages. The results are categorized based on six demonstration selection methods:

- Rand: Randomly selected demonstration samples.
- Homo-Rand: Randomly selected from the same class as the ground-truth label.
- Hetero-Rand: Randomly selected from the classes other than the ground-truth label.
- Retr: Retrieving samples based on the semantic similarity.
- Homo-Retr: Retrieving samples from the same class as the ground-truth label.
- Hetero-Retr: Retrieving samples from the classes other than the ground-truth label.

From Table 11, we can observe that retrieval improves the overall performance on all four models compared to randomly selecting demonstration. The Homo- settings result in lower scores for ChatGPT and GPT-3, but higher scores for Mistral and Llama2. In contrast, the Hetero- settings significantly degrade ICL performance.

| Model | Setting | SST2 | WNLI | RTE | MedQ | MRPC | Tweet Hate | Hate 18 | AG News | TREC | AVG |
|---|---|---|---|---|---|---|---|---|---|---|---|
| ChatGPT | Rand | 4.43% | 18.03% | 30.61% | 42.07% | 36.32% | 7.78% | -3.84% | 16.46% | 21.36% | 19.25% |
| | Homo-Rand | 2.29% | 5.63% | 19.49% | 33.28% | 29.17% | -6.50% | -6.20% | 9.90% | 36.20% | 13.70% |
| | Hetero-Rand | -13.30% | 9.86% | 8.66% | 2.79% | -4.66% | -37.10% | -27.80% | 6.40% | -20.40% | -8.39% |
| | Retr | 3.44% | 16.90% | 29.60% | 40.98% | 37.75% | 7.00% | 3.50% | 20.80% | 37.60% | 21.95% |
| | Homo-Retr | 0.34% | 4.23% | 15.88% | 34.26% | 36.27% | -2.50% | -11.80% | 33.30% | 47.00% | 17.44% |
| | Hetero-Retr | -29.82% | -16.90% | 7.58% | 2.62% | 0.49% | -39.60% | -30.60% | -25.40% | -25.60% | -17.47% |
| GPT3 | Rand | 5.64% | 22.54% | 23.90% | 15.28% | 15.74% | 11.50% | -1.90% | 12.72% | 20.64% | 14.01% |
| | Homo-Rand | 1.95% | 7.04% | 14.08% | 7.87% | 3.43% | -4.00% | -6.10% | 5.50% | 24.80% | 6.06% |
| | Hetero-Rand | 3.90% | 5.63% | -3.97% | 19.02% | 25.49% | 20.30% | 5.50% | 13.70% | 7.60% | 10.80% |
| | Retr | 4.59% | 26.76% | 22.38% | 17.38% | 13.48% | 11.10% | 4.70% | 18.20% | 31.40% | 16.67% |
| | Homo-Retr | 4.47% | -11.27% | 13.36% | 9.84% | 6.13% | 6.70% | 4.40% | 22.50% | 36.60% | 10.30% |
| | Hetero-Retr | 1.49% | 5.63% | -3.25% | 14.43% | 24.02% | 10.30% | 3.80% | -11.10% | -4.80% | 4.50% |
| Mistral | Rand | 3.67% | 7.61% | 12.27% | 3.70% | -3.63% | 1.58% | -6.22% | 2.10% | 59.08% | 8.91% |
| | Homo-Rand | 1.83% | 5.63% | 11.55% | 8.69% | 3.92% | 1.60% | -5.92% | -8.70% | 70.40% | 9.89% |
| | Hetero-Rand | -8.94% | -1.41% | -12.64% | -12.46% | -13.73% | -38.10% | 0.40% | 0.10% | 22.60% | -7.13% |
| | Retr | 8.14% | 16.90% | 14.80% | 5.74% | 3.19% | 1.00% | -1.20% | 10.90% | 79.20% | 15.41% |
| | Homo-Retr | 9.17% | 19.72% | 14.80% | 10.98% | 6.37% | 17.60% | 1.51% | 11.30% | 83.20% | 19.41% |
| | Hetero-Retr | -17.32% | -11.27% | 12.27% | -19.67% | -11.27% | -53.00% | -7.53% | -19.00% | 14.00% | -12.53% |
| Llama2 | Rand | 7.87% | 21.69% | 20.58% | 14.89% | 15.74% | - | - | 9.32% | 28.96% | 17.01% |
| | Homo-Rand | 1.15% | 8.45% | 24.19% | 18.03% | 13.97% | - | - | 5.30% | 58.00% | 18.44% |
| | Hetero-Rand | -2.18% | -14.08% | -15.16% | -7.21% | -2.21% | - | - | -1.90% | -21.80% | -9.22% |
| | Retr | 6.54% | 2.82% | 12.64% | 17.87% | 18.87% | - | - | 17.20% | 52.40% | 18.33% |
| | Homo-Retr | 10.44% | 12.68% | 22.74% | 27.70% | 23.28% | - | - | 22.80% | 70.40% | 27.15% |
| | Hetero-Retr | -11.70% | -14.08% | -12.64% | -8.52% | -2.21% | - | - | -37.10% | -20.60% | -15.26% |

Table 11: Breakdown scores of overall ICL improvement (Acc) compared to the zero-shot setting for different demonstration collection methods, corresponding to Figure 6 (a).

Table 12 and Table 13 elucidate the findings presented in Table 11.

As shown in Table 12, there is a significant increase in the discrimination factor (Retrieval vs. Random), indicating enhanced predictive ability arises substantially through the retrieved semantically-similar examples. In Homo- settings, discrimination scores are lower for ChatGPT and GPT-3 but higher for Mistral and Llama2. Conversely, in Hetero- settings, discrimination power significantly declines. Furthermore, Hetero-Retr exacerbates this decline compared to Hetero-Rand, suggesting that retrieving from non-ground-truth classes is more detrimental. This implies that LLM outputs tend to follow semantically similar demonstrations, which do not align with the ground truth in Hetero settings.

As demonstrated in Table 12, ISIF percentage decreases under Homo- settings, indicating the powers of label space and format are weakened when all demonstrations have the same label (lack diversity). Hetero- settings further reduce the ISIF percentage due to the absence of ground truth labels, which introduces uncertainty in identifying the correct label token.

| Model | Setting | SST2 | WNLI | RTE | MedQ | MRPC | Tweet Hate | Hate 18 | AG News | TREC | AVG |
|---|---|---|---|---|---|---|---|---|---|---|---|
| ChatGPT | Rand | 0.67% | 4.79% | 6.86% | 9.51% | 9.51% | -1.28% | -16.58% | -2.98% | -2.32% | 0.91% |
| | Homo-Rand | -1.26% | 5.63% | 4.69% | 5.57% | 6.13% | -12.20% | -17.00% | -4.90% | 9.20% | -0.46% |
| | Hetero-Rand | 2.06% | 8.45% | 0.00% | 16.07% | 11.27% | 1.90% | 0.60% | -7.70% | -23.40% | 1.03% |
| | Retr | -0.46% | 8.45% | 7.22% | 12.30% | 14.46% | 0.30% | -8.70% | -1.00% | 4.80% | 4.15% |
| | Homo-Retr | -0.69% | 7.04% | 0.72% | 12.13% | 15.93% | -3.10% | -6.10% | 6.20% | 12.00% | 4.90% |
| | Hetero-Retr | 0.46% | -18.31% | -1.08% | 16.07% | 13.97% | 0.20% | 0.60% | -31.40% | -28.20% | -5.30% |
| GPT3 | Rand | 1.08% | 6.20% | 6.06% | 4.23% | -2.16% | 3.08% | -8.60% | -2.98% | -3.12% | 0.42% |
| | Homo-Rand | -0.23% | 8.45% | 7.94% | -1.15% | -7.84% | -6.20% | -11.90% | -6.10% | 4.80% | -1.36% |
| | Hetero-Rand | 0.92% | 1.41% | -1.81% | 9.02% | 9.31% | 13.50% | 4.80% | -0.20% | -4.40% | 3.62% |
| | Retr | 0.92% | 18.31% | 7.58% | 6.07% | -0.98% | 4.90% | -3.30% | 0.20% | 3.20% | 4.10% |
| | Homo-Retr | 1.03% | 12.82% | 5.42% | 0.16% | -3.11% | 0.40% | -4.00% | 3.80% | 7.00% | 2.61% |
| | Hetero-Retr | -1.03% | 1.41% | -1.08% | 5.08% | 6.86% | 4.90% | 2.80% | -20.20% | -13.00% | -1.58% |
| Mistral | Rand | 1.06% | -1.97% | -1.52% | 0.62% | -5.98% | -0.36% | -6.84% | -3.20% | 1.80% | -1.82% |
| | Homo-Rand | 0.11% | 5.63% | 2.53% | 6.89% | 1.47% | 3.20% | -6.02% | -3.50% | 2.60% | 1.43% |
| | Hetero-Rand | 0.57% | -2.82% | -4.33% | -6.89% | -13.73% | -3.50% | 0.20% | -2.20% | -0.40% | -3.68% |
| | Retr | 0.92% | 8.45% | 0.36% | 3.44% | 0.74% | 3.00% | -1.71% | 1.30% | 2.20% | 2.08% |
| | Homo-Retr | 1.38% | 19.72% | 3.25% | 8.85% | 4.41% | 17.00% | 1.41% | 4.10% | 3.20% | 7.03% |
| | Hetero-Retr | -0.46% | -7.04% | -6.14% | -13.93% | -11.03% | -23.00% | -7.13% | -17.00% | -2.20% | -9.77% |
| Llama2 | Rand | -2.68% | 5.63% | 0.65% | 3.61% | 3.58% | - | - | -2.22% | -7.76% | 0.12% |
| | Homo-Rand | 1.15% | 2.82% | 4.33% | 6.23% | -4.17% | - | - | -1.40% | 1.40% | 1.48% |
| | Hetero-Rand | 0.23% | -18.31% | -18.05% | -10.16% | -5.15% | - | - | -4.20% | -22.20% | -11.12% |
| | Retr | 0.34% | -2.82% | -2.89% | 3.77% | 3.68% | - | - | 3.00% | -0.60% | 0.64% |
| | Homo-Retr | 3.21% | 1.41% | 2.17% | 12.46% | 4.90% | - | - | 9.40% | 6.20% | 5.68% |
| | Hetero-Retr | -6.54% | -18.31% | -16.25% | -11.31% | -4.90% | - | - | -33.20% | -25.00% | -16.50% |

Table 12: Breakdown scores of discrimination factor for different demonstration collection methods, corresponding to Figure 6 (b).

| Model | Setting | SST2 | WNLI | RTE | MedQ | MRPC | Tweet Hate | Hate 18 | AG News | TREC | AVG |
|---|---|---|---|---|---|---|---|---|---|---|---|
| ChatGPT | Rand | 4.43% | 19.72% | 29.96% | 24.07% | 11.76% | 10.54% | 14.08% | 23.56% | 30.00% | 18.68% |
| | Homo-Rand | 4.13% | 12.68% | 24.91% | 23.93% | 11.76% | 10.00% | 14.30% | 18.40% | 29.00% | 16.57% |
| | Hetero-Rand | -15.94% | -23.94% | 3.97% | -9.67% | -33.58% | -64.60% | -40.20% | 21.30% | 26.60% | -15.12% |
| | Retr | 4.24% | 18.31% | 29.60% | 23.28% | 11.76% | 10.10% | 14.60% | 25.60% | 34.80% | 19.14% |
| | Homo-Retr | 1.49% | 7.04% | 27.08% | 23.61% | 11.52% | 8.60% | 14.40% | 27.40% | 34.20% | 17.26% |
| | Hetero-Retr | -31.42% | 5.63% | 10.83% | -9.67% | -28.92% | -63.70% | -41.00% | 22.90% | 32.60% | -11.42% |
| GPT3 | Rand | 4.72% | 11.83% | 18.92% | 15.57% | 21.13% | 11.16% | 10.88% | 17.06% | 27.60% | 15.43% |
| | Homo-Rand | 2.75% | 2.82% | 12.64% | 14.26% | 19.36% | 7.30% | 10.40% | 14.90% | 21.20% | 11.74% |
| | Hetero-Rand | 2.87% | -8.45% | -15.16% | 14.26% | 19.12% | 4.30% | 0.30% | 17.20% | 21.80% | 6.25% |
| | Retr | 3.67% | 9.86% | 17.33% | 15.57% | 21.08% | 9.20% | 11.40% | 20.90% | 31.20% | 15.58% |
| | Homo-Retr | 4.01% | -8.45% | 14.08% | 14.92% | 19.61% | 9.10% | 11.60% | 19.20% | 29.60% | 12.63% |
| | Hetero-Retr | 2.52% | -28.17% | -15.16% | 13.93% | 19.12% | 4.70% | 0.40% | 19.00% | 20.80% | 4.13% |
| Mistral | Rand | 2.66% | 11.27% | 16.61% | 2.52% | 4.07% | 3.84% | 0.34% | 6.32% | 76.64% | 13.81% |
| | Homo-Rand | 0.34% | 4.23% | 9.39% | 0.00% | 3.19% | -11.90% | 0.50% | -5.00% | 68.20% | 7.66% |
| | Hetero-Rand | -10.21% | 0.00% | -3.61% | -5.41% | 0.49% | -37.40% | 0.30% | 3.20% | 57.40% | 0.53% |
| | Retr | 7.45% | 12.68% | 17.33% | 2.62% | 3.92% | -4.10% | 0.90% | 11.00% | 86.20% | 15.33% |
| | Homo-Retr | 6.88% | 1.41% | 11.19% | 0.98% | 2.94% | -5.70% | 0.30% | 7.70% | 80.60% | 11.81% |
| | Hetero-Retr | -17.32% | -16.90% | -6.14% | -4.43% | 0.74% | -29.60% | -0.10% | 2.80% | 65.60% | -0.59% |
| Llama2 | Rand | 8.65% | 23.38% | 26.06% | 5.44% | 4.26% | - | - | 14.74% | 60.36% | 20.41% |
| | Homo-Rand | 0.23% | 22.54% | 24.55% | 5.74% | 4.41% | - | - | 8.30% | 61.80% | 18.22% |
| | Hetero-Rand | -2.52% | 11.27% | 9.39% | 5.57% | 4.17% | - | - | 6.80% | 56.00% | 12.95% |
| | Retr | 6.77% | 22.54% | 26.35% | 5.57% | 3.68% | - | - | 16.30% | 63.00% | 20.60% |
| | Homo-Retr | 7.22% | 19.72% | 24.19% | 5.74% | 3.92% | - | - | 13.10% | 64.00% | 19.70% |
| | Hetero-Retr | -2.06% | 15.49% | -11.91% | 5.57% | 3.43% | - | - | 7.50% | 60.80% | 11.26% |

Table 13: Breakdown scores of new ISIF percentage for different demonstration collection methods, corresponding to Figure 6 (c).

# H  The instructions, inference templates and example cases of tasks

The prompt templates used in our experiments are listed in Table 14 For ChatGPT and Llama2, both instruction text and demonstrations are placed within the system message, while queries are included in the user message. In contrast, for Mistral and GPT3, which do not distinguish between system and user messages, both demonstrations and current query inputs are inputted together.

In our experiments, we observed that Llama2 often confuses demonstration samples with the actual queries, redundantly answering the demonstration questions before addressing the query. To mitigate this, we introduce "cue of demonstration" sentences in prompts. Specifically, demonstrations are enclosed with: "Following are a few demonstrations. \n {Demonstration Samples} \n End of demonstrations. Please answer the following question." This modification was not necessary for the other three models as they appropriately discern demonstrations without additional cues.

Disclaimer: Some examples in this appendix may contain offensive or disturbing content. These are extracted from the original datasets and do not represent the authors' views.

## H.1  Examples of Classification tasks

We provide two zero-shot examples and one few-shot ICL example for each tasks. Responses are categorized into OOS, ISOOF and ISIF for better understanding of their definitions. The contents within the ┆ dashed box ┆ denotes the demonstrations, only the first two demonstrations are displayed.

---

**Task**: Sentiment Analysis

**Datasets**: SST2
**Label Verbalizer**: positive, negative

Zero-shot Example

Please perform Sentiment Classification task.
Sentence: dazzling in its complexity , disturbing for its extraordinary themes , the piano teacher is a film that defies categorisation .
Label: The given sentence expresses a complex and thought-provoking opinion towards the film. OOS

Please perform Sentiment Classification task.
Sentence: it 's a charming and often affecting journey .
Label: Based on the given sentence, the label for sentiment classification would be "Positive". The word "charming" and "affecting" indicate a positive emotion or feeling towards the journey being described. ISIF

ICL Example

Please perform Sentiment Classification task.
┆ Sentence: works - mostly due to its superior cast
┆ Label: positive
┆ Sentence: own placid way
┆ Label: negative
┆ ...

Sentence: it 's a charming and often affecting journey .

Continued on next page

---

Label: positive

---

**Task**: Natural Language Inference

**Datasets**: RTE, WNLI
**Label Verbalizer**: entailment, non-entailment

Zero-shot Example

Please perform Natural Language Inference task. Given the premise and hypothesis, identify whether the premise entails the hypothesis.
Premise: The Federal Bureau of Investigation started an independent probe of the circumstances shortly after the White House made plain that President Bill Clinton considered industrial espionage a particular threat to US economic interests.
Hypothesis: A US probe of the investigation started at the instigation of the Commerce Department.
Label: the label for the inference is "Entailment Failure" or "No Entailment." ISOOF

Please perform Natural Language Inference task. Given the premise and hypothesis, identify whether the premise entails the hypothesis.
Premise: Dana Reeve, the widow of the actor Christopher Reeve, has died of lung cancer at age 44, according to the Christopher Reeve Foundation.
Hypothesis: Christopher Reeve had an accident.
Label: The premise does not entail the hypothesis. The hypothesis is about an accident involving Christopher Reeve, but the premise only mentions his death from lung cancer and does not mention any accident. ISIF

ICL Example

Please perform Natural Language Inference task. Given the premise and hypothesis, identify whether the premise entails the hypothesis.
Premise: Practically every architect of international stature has vied for the bonanza of public and private projects – including Renzo Piano, Peter Eisenman, Philip Johnson, Rafael Moneo, Helmut Jahn and Richard Rogers.
Hypothesis: Renzo Piano is an architect.
Label: entailment
Premise: Tom Online revenues for the three months ended June 30, 2005 were US$42.78 mn, an increase of 38.5% over the same period in 2004.
Hypothesis: Tom Online revenues have risen by 38.5%.
Label: entailment
...

Premise: Dana Reeve, the widow of the actor Christopher Reeve, has died of lung cancer at age 44, according to the Christopher Reeve Foundation.
Hypothesis: Christopher Reeve had an accident.
Label: non-entailment

---

**Task**: Paraphrase Detection

**Datasets**: MRPC, Medical Question Pairs
**Label Verbalizer**: equivalent, non-equivalent

Zero-shot Example

Please perform Paraphrase Detection task. Given the sentence 1 and sentence 2, please determine whether the two sentences are semantically equivalent.
Sentence 1: The pound also made progress against the dollar , reached fresh three-year highs at $ 1.6789 .

Continued on next page

Sentence2: The British pound flexed its muscle against the dollar , last up 1 percent at $ 1.6672 .
Label: The two sentences describe similar events, but they are not exactly the same. OOS

Please perform Paraphrase Detection task. Given the sentence 1 and sentence 2, please determine whether the two sentences are semantically equivalent.
Sentence 1: He said the foodservice pie business doesn 't fit the company 's long-term growth strategy .
Sentence2: " The foodservice pie business does not fit our long-term growth strategy .
Label: The two sentences are semantically equivalent. Both sentences express that the foodservice pie business is not in line with the company's long-term growth strategy. ISIF

| ICL Example |
| --- |

Please perform Paraphrase Detection task. Given the sentence 1 and sentence 2, please determine whether the two sentences are semantically equivalent.
Sentence 1: Hoffa , 62 , vanished on the afternoon of July 30 , 1975 , from a Bloomfield Township parking lot in Oakland County , about 25 miles north of Detroit .
Sentence 2: Hoffa , 62 , vanished on the afternoon of July 30 , 1975 , from a parking lot in a Detroit suburb in Oakland County .
Label: equivalent
Sentence 1: " We are very pleased that the court dismissed the vast majority of the plaintiff 's claims , " Coke spokeswoman Sonya Soutus said .
Sentence 2: " We are very pleased that the court dismissed the vast majority of the plaintiff 's claims , " said Coke spokesman Ben Deutsch .
Label: equivalent
...

Sentence 1: He said the foodservice pie business doesn 't fit the company 's long-term growth strategy .
Sentence2: " The foodservice pie business does not fit our long-term growth strategy .
Label: equivalent

| **Task**: Hate Speech Detection |
| --- |
| **Datasets**: Tweet Hate, Hate Speech 18
**Label Verbalizer**: hate, non-hate |
| Zero-shot Example |

Please perform Hate Speech Detection task.
Sentence: This is Principal Max Caulfield. Let me be emphatic. I have an enormous skank inserted into my butt. Thank you.
Label: This sentence contains crude and vulgar language that can be considered offensive and disrespectful. The label for this sentence should be "Hate Speech". ISOOF

Please perform Hate Speech Detection task.
Sentence: @user @user If book Claire wanted to "stay in the kitchen cooking for her man," she would've stayed in Inverness while Frank did his research, and the Outlander series would've ended up being less than two chapters long. Perhaps that's what you want, but I prefer the story that we did get.
Label: Neutral. Explanation: The sentence expresses a preference for the way the Outlander series developed. OOS

| ICL Example |
| --- |
| **Sample Prompt Template**: |

Continued on next page

Please perform Hate Speech Detection task.

Sentence: @user Women as 'sacred' is the tell that Kelly views women thru a Madonna/whore lens (read: misogyny) - and he's racist to boot
Label: non-hate
Sentence: @user Best solution to bring permanent peace in Valley is to encourage migration of Sikhs,Jats&Rajputs to Valley,set up business
Label: non-hate
...

Sentence: @user @user If book Claire wanted to "stay in the kitchen cooking for her man," she would've stayed in Inverness while Frank did his research, and the Outlander series would've ended up being less than two chapters long. Perhaps that's what you want, but I prefer the story that we did get.
Label: non-hate

---

**Task**: Multiclass Classification

**Datasets**: AG News
**Label Verbalizer**: world, sports, business, sciencetechnology

Zero-shot Example

Please perform Sentence Classification task. Given the sentence, please identify which news type does this sentence of news article belong to.
Sentence: Kerry Says He's in a 'Fighting Mood' (AP) AP - Democratic Sen. John Kerry said Saturday he's in "fighting mood" with two months to go to the presidential as his allies defended him from questions about his valor in Vietnam.
Label: This sentence belongs to the "Politics" news type. OOS

Please perform Sentence Classification task. Given the sentence, please identify which news type does this sentence of news article belong to.
Sentence: Some People Not Eligible to Get in on Google IPO Google has billed its IPO as a way for everyday people to get in on the process, denying Wall Street the usual stranglehold it's had on IPOs. Public bidding, a minimum of just five shares, an open process with 28 underwriters - all this pointed to a new level of public participation. But this isn't the case.
Label: Business and Economy (specifically, Technology and IPOs) ISIF

ICL Example

Please perform Sentence Classification task. Given the sentences, please identify which type of news article do they belong to.

Sentence: AOL Moves Beyond Passwords for Log-Ons (AP) AP - Passwords alone won't be enough to get onto America Online under a new, optional log-on service that makes AOL the first major U.S. online business to offer customers a second layer of security.
Label: science/technology
Sentence: Stampede kills five at railway station in Indian capital; seven injured (Canadian Press) Canadian Press - NEW DELHI (AP) - A stampede at the main railway station in India's capital killed at least five people and injured seven on Saturday, the railway minister said.
Label: world
...

Sentence: Some People Not Eligible to Get in on Google IPO Google has billed its IPO as a way for everyday people to get in on the process, denying Wall Street the usual stranglehold it's had on IPOs. Public bidding, a minimum of just five shares, an open process with 28 underwriters - all this pointed to a new level of public participation. But this isn't the case.

Continued on next page

Label: science/technology

---

**Task**: Multiclass Classification

---

**Datasets**: TREC
**Label Verbalizer**: abbreviation, entity, description and abstract concept, human being, location, numerical value

Zero-shot Example

---

Please perform Sentence Classification task. Given the question, please identify the type of this question.
Sentence: What county is Modesto , California in ?
Label: This question is a Factoid or Exact Answer type of question. The goal is to identify the specific fact or answer from the given sentence. In this case, the fact being sought is the name of the county that Modesto, California is located in. OOS

Please perform Sentence Classification task. Given the question, please identify the type of this question.
Sentence: How far is it from Denver to Aspen ?
Label: The question you provided is asking for information, specifically asking for the distance between Denver and Aspen. Therefore, the label for this sentence would be: Label: Question (Q) - Information Request (IR) OOS

ICL Example

---

Please perform Sentence Classification task. Given the question, please identify the type of this question

> Sentence: What are the short- and long-term effects of underage drinking ?
> Label: description and abstract concept
> Sentence: What are the short- and long-term effects of underage drinking ?
> Label: description and abstract concept
> ...

Sentence: How far is it from Denver to Aspen ?
Label: numerical value

---

Table 14: Label Verbalizers and sample Prompt Templates for classification datasets. The contents within the ┊ dashed box ┊ denotes the demonstrations, the words in purple denotes models' answer and the category of OOS/ISOOF/ISIF is marked in orange. Only the first two demonstrations are displayed.

## H.2 Examples of Sequence Labelling tasks

---

**Task**: NER

---

**Datasets**: CoNLL 2003
**Types**: Person, Location, Organization, Miscellaneous

Zero-shot Example

---

Please perform Named Entity Recognition task. Given the sentence, please identify all named entities and classify their types.
Sentence: Japan began the defence of their Asian Cup title with a lucky 2-1 win against Syria in a Group C championship match on Friday .

Continued on next page

---

Entities: 1. Japan (ORGANIZATION, Country) 2. Asian Cup (EVENT) 3. title (EVENT, specifically the title that Japan is defending) 4. Syria (ORGANIZATION, Country) 5. Group C (ORGANIZATION, Soccer Group) 6. championship match (EVENT, Soccer Match) 7. Friday (DATE)

| ICL Example |
| --- |

Please perform Named Entity Recognition task. Given the sentence, please identify all named entities and classify their types.

Sentence: The defending world champion Atlanta Braves , with the best record and best pitching in baseball , added another weapon Wednesday , acquiring Denny Neagle , the winningest left-hander in the National League , from the Pittsburgh Pirates .
Entities:
Person: Denny Neagle
Location: None
Organization: Atlanta Braves, Pittsburgh Pirates
Miscellaneous: National League
...

Sentence: Japan began the defence of their Asian Cup title with a lucky 2-1 win against Syria in a Group C championship match on Friday .
Entities:
Person: None
Location: Japan, Syria
Organization: None
Miscellaneous: Asian Cup, Group C, championship

| **Task**: NER |
| --- |

**Datasets**: WNUT 2017
**Types**: Person, Location, Corporation, Product, Creative-work, Group

| Zero-shot Example |
| --- |

Please perform Named Entity Recognition task. Given the sentence, please identify all named entities and classify their types.
Sentence: "Arrangements are in place to carry the mortal remains of the martyrs to their native places immediately after weather becomes clear , " Defence Spokesman Colonel Rajesh Kalia said .
Entities: 1. Defence Spokesman: PERSON (Organization Role) 2. Colonel Rajesh Kalia: PERSON 3. martyrs: PERSON (Mass Noun) 4. native places: LOCATION

| ICL Example |
| --- |

Please perform Named Entity Recognition task. Given the sentence, please identify all named entities and classify their types.

Sentence: @paulwalk It 's the view from where I 'm living for two weeks . Empire State Building = ESB . Pretty bad storm here last evening .
Entities:
Corporation: None
Creative-work: None
Group: None
Location: Empire State Building, ESB
Person: None
Product: None
...

Sentence: & gt ; * The soldier was killed when another avalanche hit an army barracks in the northern area of Sonmarg , said a military spokesman .
Entities:

Corporation: None
Creative-work: None
Group: None
Location: Sonmarg
Person: a military spokesperson
Event: None
Product: None

---

**Task**: ABSA

**Datasets**: SemEval 2014, 2015, 2016
**Types**: positive, negative, neutral, conflict* (Type "conflict" was only annotated in SemEval2014)

Zero-shot Example

Please perform Aspect Based Sentiment Analysis task. Given a sentence, please extract all aspect terms and the sentiment that the author is expressing towards them.
Sentence: I have to say they have one of the fastest delivery times in the city.
Label:
Aspect Terms: delivery times
Sentiment: Positive. The author is expressing a positive sentiment towards the aspect term "delivery times" by stating that they are among the fastest in the city.

ICL Example

Please perform Aspect Based Sentiment Analysis task. Given a sentence, please extract all aspect terms and the sentiment that the author is expressing towards them.
Sentence: While there's a decent menu, it shouldn't take ten minutes to get your drinks and 45 for a dessert pizza.
Label:
Aspect Term: menu, Sentiment: positive
Aspect Term: drinks, Sentiment: neutral
Aspect Term: dessert pizza, Sentiment: neutral
...

Sentence: Certainly not the best sushi in New York, however, it is always fresh, and the place is very clean, sterile.
Label:
Aspect Term: sushi, Sentiment: neutral (or mixed, as the author mentions it's not the best but also mentions it's always fresh)
Aspect Term: place, Sentiment: positive

---

Table 15: Sample prompt template for sequence labelling datasets. The contents within the dashed box denotes the demonstrations, and the words in purple denotes models' answer. Only the first demonstration is displayed as example, for clear presentation.

### H.3 Examples and Case Study of Generation tasks

**Case study of text generation tasks**

For Reddit dataset, both the source texts and the ground truth summaries are in casual style. In a zero-shot setting, a well-aligned LLM like ChatGPT defaults to a formal tone if no specific instruction on tone is provided. With ICL, the summary aligns more with the provided demonstrations, adopting short verb phrases separated by commas (e.g., "got drunk, went to their favorite bar..."). However, it retains some formal discourse elements.

For the SamSum dataset, ICL prompts the model to adopt a writing style characterized by short sentences in the "{Person Name} {actions}" structure. Each short sentence in the response begins with a person's name, contrasting with the longer sentences typically

generated in a zero-shot setting. The model also adapts to similar vocabulary in a formal setting.

In the ROCStories dataset, the ground truth always comprises exactly four sentences. The model follows this pattern with ICL, though the generated sentences tend to be longer than the ground truth. When provided with style-edited demonstrations that deviate from the four-sentence rule, the model's responses also vary in the number of sentences.

---

**Task**: Summarizaton

**Dataset**: Reddit

Zero-shot Example

Please summarize the given forum post.
Text:
so this happened the early hours of this morning , me and my friends went out for a few drinks and i had every intention of going home early and not getting drunk . problem was i completely forgot that i was a responsible adult and continued drinking . i suggested we go to my favourite bar it was reasonably early so we would get there and not have to queue . once we got over to the bar everything was going great , free entry followed by a night of free drinks and a good bit of dancing . in any case it got to the time where i needed to break the seal , unable to walk in a straight line i staggered into the toilet . tripped and on my way down grabbed the only thing that i could put my hands on . the pipe that connected the toilet to the top of it where the water goes into it . immediately water started pissing out everywhere . and me being the proficient tactician i am , decided that i could fix it . this was a bad idea , as something that could easily be fixed by placing a bucket . no no i was going to take it lying down . by the power of my bare hands i was going to fix the pipes back together . obviously i 'm not superman and the toilet completely burst . i 'm now covered in water . the water is pissing out the toilet everywhere .
Summary:
The author shares a humorous story about a night out with friends that ended in a chaotic situation involving a broken toilet pipe in a bar.

ICL Example

Please summarize the given forum post.
Text:
so , a job opened up on one of the other teams in my department , which also happens to be overseen by my manager . i was one of the people interviewed for the position , as was someone else on my team . today , boss sits down with me : " i decided to give the position to ( other team member ) . you 're my hardest-working employee . we already have 1 person on maternity leave , and when ( other team member ) goes to new position , i need someone i can count on to make sure everything runs smoothly during the transition . " the frustrating part is knowing that she actually means it . my boss is someone who is unable to not say exactly what is on her mind . i 'd be way happier being told " you were a good choice , but do n't quite have what we 're looking for . " instead of " i think you 're the better fit , but i just ca n't spare you . " fortunately , it sounds like that team may be expanding soon , so i might have a second shot at the brass ring .", "target": "missed out on a job by being too useful in my current job .
Summary:
missed out on a job by being too useful in my current job .
...

Text:
so this happened the early hours of this morning , me and my friends went out for a few drinks ... (Same paragraph as zero-shot sample)

Continued on next page

---

Summary:

The person got drunk, went to their favorite bar, accidentally broke the toilet pipe, and attempted to fix it with their bare hands, resulting in a flooded restroom.

| ICL Example - Active |
| --- |

Please summarize the given forum post.

Text:
so , a job opened up on one of the other teams in my department , which also happens to be overseen by my manager ... (Same demonstration paragraph as base ICL sample)
Summary:
I missed the job opportunity because I was too useful in my current role.
...

Text:
so this happened the early hours of this morning , me and my friends went out for a few drinks ... (Same paragraph as zero-shot sample)
Summary:
The user went out drinking with friends and ended up causing a toilet to burst by trying to fix it with their bare hands, leading to a messy and embarrassing situation.

| ICL Example - Formal |
| --- |

Please summarize the given forum post.

Text:
so , a job opened up on one of the other teams in my department , which also happens to be overseen by my manager... (Same demonstration paragraph as base ICL sample)
Summary:
I failed to secure a job opportunity due to my exceptional performance in my current role.
...

Text:
so this happened the early hours of this morning , me and my friends went out for a few drinks ... (Same paragraph as zero-shot sample)
Summary:
While at a bar, the author drunkenly tried to fix a broken toilet, but only made the situation much worse by causing it to burst and getting themselves covered in water.

| ICL Example - Passive |
| --- |

Please summarize the given forum post.

Text:
so , a job opened up on one of the other teams in my department , which also happens to be overseen by my manager ... (Same demonstration paragraph as base ICL sample)
Summary:
By being too useful in my current job, a job was missed out.
...

Text:
so this happened the early hours of this morning , me and my friends went out for a few drinks ... (Same paragraph as zero-shot sample)
Summary:
A night out with friends ended in a disastrous attempt to fix a broken toilet pipe in a bar, resulting in water spraying everywhere and leaving the person covered in water.

| **Task**: Summarizaton |
| --- |

**Dataset**: SamSum

| Zero-shot Example |
| --- |

Please summarize the given dialogue.

*Continued on next page*

Text:
Ivan: hey eric
Eric: yeah man
Ivan: so youre coming to the wedding
Eric: your brother's
Ivan: yea
Eric: i dont know mannn
Ivan: YOU DONT KNOW??
Eric: i just have a lot to do at home, plus i dont know if my parents would let me
Ivan: ill take care of your parents
Eric: youre telling me you have the guts to talk to them XD
Ivan: thats my problem
Eric: okay man, if you say so
Ivan: yea just be there
Eric: alright"
Summary:
Ivan invites Eric to his brother's wedding, but Eric is unsure if he can attend due to responsibilities at home and potential parental restrictions. Ivan offers to talk to Eric's parents, but Eric is doubtful. Ultimately, Eric agrees to try to attend the wedding.

---

ICL Example

---

Please summarize the given dialogue.

Text:
Chad: hey, u finished for today?
Logan: no, i have one meeting left in the afternoon, at 2pm
Chad: easy day ;)
Logan: well yeah but i'm just so fucking tired today
Logan: i just couldn't get out of bed
Chad: yeah same here
Logan: i guess i'll be heading back to bed in a minute
Chad: ⟨file_photo⟩
Logan: hahaa
Logan: so i'm not the only one with such plans;D
Chad: exactly ;D i just need to get some more tea
Logan: i can scan you mine and e-mail it to you xD
Chad: ;D
Summary:
Logan has one more meeting at 2 pm. Chad and Logan are both tired and will spend their day in beds.
...

Text:
Ivan: hey eric
Eric: yeah man
Ivan: so youre coming to the wedding
Eric: your brother's ... (Same paragraph as zero-shot sample)
Summary:
Ivan invites Eric to his brother's wedding, but Eric is unsure if he can attend due to various reasons. Ivan offers to take care of talking to Eric's parents about it. They agree that Eric will try to make it to the wedding.

---

ICL Example - Active

---

Please summarize the given dialogue.

Text:
Chad: hey, u finished for today?
Logan: no, i have one meeting left in the afternoon, at 2pm... (Same demonstration paragraph as base ICL sample)
Summary:
Logan will attend one more meeting at 2 pm. Chad and Logan, both tired, will spend their day in bed.
...

Text:
Ivan: hey eric
Eric: yeah man
Ivan: so youre coming to the wedding
Eric: your brother's ... (Same paragraph as zero-shot sample)
Summary:
Ivan invites Eric to his brother's wedding, Eric is unsure due to tasks at home and uncertain if his parents would allow it. Ivan offers to handle the parental conversation. Eric agrees to attend the wedding upon Ivan's assurance.

|ICL Example - Formal|

Please summarize the given dialogue.

Text:
Chad: hey, u finished for today?
Logan: no, i have one meeting left in the afternoon, at 2pm ... (Same demonstration paragraph as base ICL sample)
Summary:
Logan is scheduled for an additional meeting at 2 pm. Both Chad and Logan are fatigued and intend to rest in their beds for the remainder of the day.
...

Text:
Ivan: hey eric
Eric: yeah man
Ivan: so youre coming to the wedding
Eric: your brother's ... (Same paragraph as zero-shot sample)
Summary:
Ivan invites Eric to his brother's wedding, but Eric is unsure if he can attend due to household responsibilities and parental permission. Ivan offers to speak to Eric's parents about it. Despite Eric's skepticism, he agrees to try and attend the wedding.

|ICL Example - Passive|

Please summarize the given dialogue.

Text:
Chad: hey, u finished for today?
Logan: no, i have one meeting left in the afternoon, at 2pm ... (Same demonstration paragraph as base ICL sample)
Summary:
One more meeting is scheduled for Logan at 2 pm. Chad and Logan are both tired, they will spend their day in beds.
...

Text:
Ivan: hey eric
Eric: yeah man
Ivan: so youre coming to the wedding
Eric: your brother's ... (Same paragraph as zero-shot sample)
Summary:

Ivan invites Eric to his brother's wedding, but Eric is unsure if he can attend due to responsibilities at home and concerns about his parents. Ivan offers to talk to Eric's parents, and although Eric doubts Ivan's ability, he agrees to try and make it to the wedding.

---

**Task**: Story Generation
**Dataset**: ROCStories, ROCStories Ending

### Zero-shot Example

Given an incomplete story, please generate a reasonable ending for the story. Please finish the ending in one single sentence. (This instructions is only included for ROCStories Ending subset.)
Incomplete Story:
Louisa and her family took a trip to Epcot.
Story Ending:
Louisa and her family took a trip to Epcot, filled with excitement and anticipation. They started their day by exploring the different pavilions, marveling at the unique architecture and immersive experiences that each country showcased. From sampling authentic cuisine in France to watching a traditional Japanese drum performance, they were fully immersed in the diverse cultures represented at Epcot.
As the day went on, they found themselves in Future World, where they rode thrilling attractions like Test Track and Soarin'. The highlight of their day was watching the fireworks show over the iconic Spaceship Earth, its lights dancing in the night sky. Louisa held hands with her parents and siblings, feeling grateful for this special moment together.
As they made their way back to the entrance, Louisa couldn't help but reflect on the magical day they had just experienced. Epcot had truly captured their hearts, leaving them with memories that would last a lifetime. Louisa knew that this trip was one they would never forget, a day of joy, laughter, and a deep appreciation for the world's beauty.

### ICL Example

Incomplete Story:
Jim invited his friends to his house on Friday.
Story Ending:
When Friday arrived, Jim totally forgot about his plans. However, he decided to go out to eat. His friends called him since they couldn't get into the house. Jim ignored the messages since he totally forgot about his plans.

...

Incomplete Story:
Louisa and her family took a trip to Epcot.
Story Ending:
They were all excited to experience the different countries and attractions at the park. As they walked through the entrance gates, Louisa's little brother ran off towards the Spaceship Earth ride. They spent the whole day exploring the park, trying out different foods, taking pictures with Disney characters, and watching the fireworks show in the evening. Louisa couldn't stop smiling the entire time, grateful for the opportunity to create unforgettable memories with her family at Epcot.

### ICL Example - Active

Continued on next page

Incomplete Story:
Jim invited his friends to his house on Friday.
Story Ending:
On Friday, Jim completely forgot about his plans and decided to go out to eat, ignoring messages from his friends who couldn't get into the house.

...

Incomplete Story:
Louisa and her family took a trip to Epcot.
Story Ending:
Once they arrived at Epcot, Louisa and her family were amazed by the vibrant atmosphere and diverse cultures represented throughout the park. They started their adventure by exploring the different pavilions, trying out delicious snacks from around the world, and experiencing thrilling rides. As the day went on, Louisa felt a sense of joy and unity being surrounded by her loved ones in such a magical place. The family ended the day by watching the spectacular fireworks show, creating memories that would last a lifetime. Louisa couldn't help but feel grateful for this unforgettable trip to Epcot with her family.

ICL Example - Formal

Incomplete Story:
Jim invited his friends to his house on Friday.
Story Ending:
Upon the arrival of Friday, Jim completely overlooked his plans but opted to dine out. His acquaintances attempted to reach him as they were unable to gain access to the residence. Jim disregarded the notifications, as his plans had slipped from his memory.

...

Incomplete Story:
Louisa and her family took a trip to Epcot.
Story Ending:
As they wandered through the park, Louisa's younger sister, Emma, spotted her favorite Disney character, Mickey Mouse. Excitedly, she ran towards him, causing a commotion. Louisa and her parents had to chase after her, laughing and enjoying the chaos of their Epcot adventure. Despite the unexpected detour, it turned out to be a memorable and joyful day for the whole family.

ICL Example - Passive

Incomplete Story:
Jim invited his friends to his house on Friday.
Story Ending:
When Friday arrived, Jim's plans were totally forgotten by himself. However, the decision to go out to eat was made. He was called by his friends as they couldn't get into his house, but the messages was ignored by him as he forgot about his plans.

...

Incomplete Story:
Louisa and her family took a trip to Epcot.
Story Ending:
They were all thrilled to explore the different countries in the World Showcase. Louisa's favorite was Italy, where they indulged in delicious gelato. As the day went on, they rode thrilling rides and watched captivating performances. The night ended with a spectacular fireworks show that left them in awe. Louisa and her family returned to their hotel, exhausted but grateful for the unforgettable day they had at Epcot.

Table 16: Sample prompt template for generation datasets. The contents within the dashed box denotes the demonstrations, and the words in purple denotes models' answer. The words in teal colour indicates the sign of style shifts. Only the first demonstration is displayed as example, for clear presentation.

