# OpenReview forum: "Does In-Context Learning Really Learn? Rethinking How Large Language Models Respond and Solve Tasks via In-Context Learning"
_colmweb.org/COLM/2024/Conference — COLM_

### Official Review · Reviewer_aa3n · 2024-05-08

**Rating:** 7
**Confidence:** 4
**Ethics Flag:** 1

**Summary:**

The paper hypothesizes that a lot of improvements from in-context learning (ICL) can be attributed to models following the label space and format better. The paper conducts experiments to breakdown ICL improvements along three dimensions – label space, label format and discrimination to show that ICL mostly helps due to the former two. The paper also shows that ICL provides similar benefits to instructions for following the label space and format. All of these are also studied under a setting where the few-shot examples are retrieved based on test input, instead of a random set of few-shot examples.

**Questions To Authors:**

1. Figure 2 (and 3) — since demonstrations are randomly sampled, it would be good to show std error / confidence intervals.

2. Sec 6.2 – an interesting experiment to add here would be retrieve most similar examples from a different label than y^\prime. How does that affect performance? (I believe both the proposed controlled experiments, ‘cheat-random’ and ‘cheat-retrieval’ sample from the same class y^\prime)

3. In the intro, it was not really clear what you mean by ‘discrimination’ as the third factor of improvement. This did become in sec 3.2 though. I think the writing earlier could be improved to make it more clearer (is it correct to say that any improvements ICL provides beyond just label space and format are part of ‘discrimination’).

4. Writing – the research questions and takeaways at the end of the introduction seemed like a mix of results and were not very clear e.g. where does ‘demonstration style’ come in? Also for the ‘mechanism of retrieval’ it would be good to connect that back to the three sources of improvement. I think the writing in this part can definitely be improved!

**Reasons To Accept:**

1. The paper has some findings which are pretty interesting and also would be practically useful when using ICL.

   a. Figure 3 – the result that after ICL there are a lot of examples within ISIF (i.e. label format and space is correct) where prediction changes from right to wrong is quite interesting! Do you have a guess for why that occurs?

   b.  Figure 4 – the finding that instructions help as much as ICL because the improvements for ICL come from label space and format is also interesting!

    c. The section 5.3 finding is also interesting. We have seen in prior works that ICL w/ randomized labels performs quite well. Figure 5 shows that this is because models are able to maintain the benefit from label space and format. And whenever performance does drop, it is due to the discriminative power being affected with random labels.

2. The experiments are exhaustive — 9 classification datasets, as well as 4 generation datasets. The models used are also a reasonable mix of best models from closed and open source models.

**Reasons To Reject:**

1. All the experiments are conducted in a setting where you have first fixed how you evaluate models (using the post processing script). But I think there are alternate ways to evaluate (e.g. compute probability of options, or use LLMs to evaluate) which will not have the issue of requiring a strict format for evaluation. E.g. GPT-4 might be able to judge correctness even if they are OOS or ISOOF and not following strict format.

2. All of the results rely on being able to classify the response as OOS, ISOOF and ISIF. While ISIF is fine, it’s not clear or mentioned how we can distinguish between OOS and ISOOF automatically. More importantly, I think the definitions of the three categories can be made much more precise (right now, it seems vague and there are only some examples to how they are defined).

3. While the specific details of experiments are different between this work and Min et al. 2022 and Pan et al, 2023, I think the conclusions/takeaways which this paper finds can also be inferred from those two works. Specifically, they show that ICL with randomized label assignment does well — under this randomization, the only thing the model might be learning is taks recognition, and the format & space of the labels.

---

> ### Author Rebuttal · Authors · 2024-05-31
>
> ### Response to Reviewer aa3n (1/3)
>
> Thanks for your valuable comments, helpful suggestions and appreciation of our work!
>
> > **Q1: Why a lot of examples within the ISIF change from right to wrong?**
>
> Thank you for raising this valuable question. Our conjecture is that the quality of demonstrations significantly impacts ICL performance, and random examples may be unrelated or even detrimental to the prediction of some instances. To test this, we conduct another set of experiments on the retrieval setting where ICL demonstrations are selected based on the retriever. It can be observed that the R2W rate can be moderately alleviated by retrieved demonstrations, as the following table shows:
>
> |         | Random  |        |         |        | Retrieved |        |         |        |
> |---------|---------|--------|---------|--------|-----------|--------|---------|--------|
> |         | ChatGPT | GPT3   | Mistral | Llama2 | ChatGPT   | GPT3   | Mistral | Llama2 |
> | R2R     | 60.40%  | 65.67% | 71.23%  | 63.77% | 61.10%    | 65.01% | 71.80%  | 65.13% |
> | W2W     | 16.10%  | 15.99% | 14.64%  | 10.29% | 13.15%    | 12.72% | 10.96%  | 10.24% |
> | W2R     | 13.13%  | 10.34% | 7.25%   | 12.47% | 15.66%    | 13.72% | 10.97%  | 12.47% |
> | R2W     | 10.37%  | 7.96%  | 6.88%   | 13.40% | 10.09%    | 8.55%  | 6.26%   | 12.16% |
> | W2R-R2W | 2.76%   | 2.38%  | 0.37%   | -0.93% | 5.57%     | 5.18%  | 4.71%   | 0.32%  |
>
> Results are averaged across 9 classification datasets
>
> From the table above, we observe an improvement in the differences between W2R and R2W rate for all models utilizing retrieved demonstrations. However, the R2W rate remains significant even with retrieved semantically-similar demonstrations. Potential explanations include: 1) the retrieval method based on semantic similarity is imperfect; 2) demonstrations can be regarded as "additional parameters" that influence the token generation probability distribution. We plan to explore this aspect in future work.

---

> > ### Author Response · Authors · 2024-05-31
> >
> > ### Response to Reviewer aa3n (2/3)
> >
> > > **W1: All the experiments are conducted via the post-processing, GPT-4 evaluation and probability evaluation can be adopted.**
> >
> > Thanks for your suggestion. We understand that the GPT-4 evaluation can overlook some format problems and could be more robust than the post-processing script. However, in this paper, we want to thoroughly explore the precise contributions (label space, format, or the discrimination) of demonstrations towards improving end-task performance. Therefore, we have to rely on post-processing to categorize the instances according to the label space and format. Moreover, evaluations based on LLMs are not guaranteed to be accurate, and hence could not provide a reliable conclusion.
> >
> > > **W2: The definition of OOS, ISOOF and ISIF is not clear.**
> >
> > Thanks for your suggestion, we will take Reviewer tah4’s suggestions and use the simpler but more clear definition: Label Space := all acceptable labels regardless of
> > synonyms. Label Format := exact string match.
> >
> > > **W3: Conclusions and takeaways can be inferred from previous works.**
> >
> > Thank you for your insightful comment. We respectfully disagree with the claim that our conclusions/takeaways could be inferred from previous works. We take a deep look into the factors contributing to ICL, attempting to provide a definitive answer regarding how ICL benefits model prediction. Our finding can address some unanswered findings including the phenomenon that demonstrations with incorrect labels still enable the model to perform well.
> > Specifically,
> >
> > (1) Previous studies have relied on the assumption that "incorrect labels can still serve as valid demonstrations," yet the underlying reasons for the effectiveness of ICL remain unexplored. These studies lack a quantitative analysis of the specific factors in incorrect labels that benefit ICL, and their experiments using randomized labels within demonstrations do not provide sufficient insights.
> >
> > (2) The usage of term "Format" differs. As we stated in Appendix E, previous works' analytical scope on “format” can be interpreted as how format of demonstrations (query-answer pairing pattern) affect performance. In contrast, we specifically decompose the factors into format, space and discrimination, and identify the regulation effect brought by ICL.
> > We find that a significant portion of ICL's contributions stem from the label space and the format's regulatory effect on model responses, which is absent in previous works.
> >
> > (3) Pan et. al. 2023 employ the terms of "Task Recognition" (TR) and "Task Learning" (TL) as two factors contributing to ICL performance. TR denotes the model can maintain good performance even with incorrect input-label mappings, this resembles previous work Min et al.
> > (2022). TL can indeed be conceptualized as label space regulation in our paper. Their experiments on TL are are limited to altering the label space (such as converting "positive"/"negative" labels to 0/1 or other symbols).
> > However, since the models they adopted are not instruction-tuned, they wouldn’t be able to explore the regulation effect on response format. This is one major difference between our work and these previous works, as the response generated by current general-purpose, human-instruction-aligned LLMs differ from the early models.

---

> > ### Author Response · Authors · 2024-05-31
> >
> > ### Response to Reviewer aa3n (3/3)
> >
> > > **Q2: Show std error/ confidence intervals for results**
> >
> > Thank you for the advice. We will amend the tables with std error in the final version.
> >
> >
> > > **Q3: An interesting experiment to add here would be retrieve most similar examples from a different label**
> >
> >
> > Thank you for the suggestion of such a valuable ablation experiment. Follow your suggestion, we conduct experiment with demonstrations retrieved/randomly sampled from the collection of $\neg\ y^\prime$, in comparison with Section 6.2, and call them "reverse-cheat-rand" and "reverse-cheat-retrieval". The results are listed below (For length limit, we only put the scores averaged across datasets here).
> >
> > C- stands for cheat, RC- stands for reverse-cheat
> > | ICL Improvement | Rand   | C-Rand | RC-Rand | Retrieve | C-Retrieve | RC-Retrieve |
> > |-----------------|--------|------------|--------------|----------|----------------|-------------------|
> > | ChatGPT         | 19.25% |     13.70% |       -8.39% |   21.95% |         17.44% |           -17.47% |
> > | GPT3            | 14.01% |      6.06% |       10.80% |   16.67% |         10.30% |             4.50% |
> > | Mistral         |  8.91% |      9.89% |       -7.13% |   15.41% |         19.41% |           -12.53% |
> > | Llama2          | 17.01% |     18.44% |       -9.22% |   18.33% |         27.15% |           -15.26% |
> >
> > | Discrimination &nbsp;&nbsp;&nbsp; | Rand   | C-Rand | RC-Rand | Retrieve | C-Retrieve | RC-Retrieve |
> > |----------------|--------|------------|--------------|----------|----------------|-------------------|
> > | ChatGPT        |  0.91% |     -0.46% |        1.03% |    4.15% |          4.90% |            -5.30% |
> > | GPT3           |  0.42% |     -1.36% |        3.62% |    4.10% |          2.61% |            -1.58% |
> > | Mistral        | -1.82% |      1.43% |       -3.68% |    2.08% |          7.03% |            -9.77% |
> > | Llama2         |  0.12% |      1.48% |      -11.12% |    0.64% |          5.68% |           -16.50% |
> >
> > | New ISIF Rate &nbsp;&nbsp;&nbsp;&nbsp;| Rand   | C-Rand | RC-Rand | Retrieve | C-retrieve | RC-Retrieve |
> > |----------|--------|------------|--------------|----------|----------------|-------------------|
> > | ChatGPT  | 18.68% |     16.57% |      -15.12% |   19.14% |         17.26% |           -11.42% |
> > | GPT3     | 15.43% |     11.74% |        6.25% |   15.58% |         12.63% |             4.13% |
> > | Mistral  | 13.81% |      7.66% |        0.53% |   15.33% |         11.81% |            -0.59% |
> > | Llama2   | 20.41% |     18.22% |       12.95% |   20.60% |         19.70% |            11.26% |
> >
> > It is obvious that the reverse-cheat settings significantly degrade performance, impacting both discrimination power and the regulation of label space and format (new ISIF). For the decrease in ISIF, as most of our tasks are binary classification tasks, failing to be exposed with ground truth label will make the model blind in deciding the label token to output, supposing the model could still discriminate the class correctly.
> > The decrease of new ISIF is especially prominent for ChatGPT, while the other three models are having similar level of decrease. In the meantime, we find that the discrimination power is also compromised, and the decrease in discrimination scores are more severe in the two open-sourced models. Aligning with the analysis in Section 6.2, Mistral and Llama2 appear more susceptible to following labels in demonstrations when the queries are semantically similar.
> >
> >
> > > **Q4 Writing suggestions**
> >
> > Thank you for the writing suggestions. We will improve the introduction section and revise the summary of takeaways based on your suggestions for the camera-ready version.
> >
> > Thanks for your appreciation again!

---

> > > ### Comment · Reviewer_aa3n · 2024-06-04
> > > **Response to rebuttal**
> > >
> > > I thank the authors for the detailed rebuttal! I think a lot of concerns that I had seem to be addressed (e.g. definition of OOS , ISOOF) and hope they are incorporated into the next version of the draft. Regarding the concern of novelty, it seems like the authors do agree with some similarities to Pan et al 2023 but point out two main differences — more fine-grained treatment of what we mean by ‘format’ and using instruction-tuned models, both of which seem fair. Given this, I would be inclined to increase the score.

---

### Official Review · Reviewer_xiZm · 2024-05-11

**Rating:** 5
**Confidence:** 3
**Ethics Flag:** 1

**Summary:**

This paper aims to study the mechanism of why in-context learning works, and it decomposes the overall ICL performance into label space, format, and discrimination. They experiment with four LLMs (ChatGPT, GPT-3, LLaMA 2, and Mistral), and experiment on nine classification datasets and four generation datasets. They show that label space and format adjustments are critical to ICL's efficacy, while discrimination improvements were marginal.

**Reasons To Accept:**

This paper decomposes the ICL performance into label space, label format, and discrimination, offering a structured analysis method. They also experiment with multiple models and tasks; the results are solid and robust. The discussion regarding the generation tasks is interesting.

**Reasons To Reject:**

While this paper discusses label space and format, it would benefit from an expanded discussion on related work that emphasizes the importance of them, as a series of recent works have specified that the formats and domains are important. For example, some conclusions of this paper are already discussed in "Z-ICL: Zero-Shot In-Context Learning with Pseudo-Demonstrations", which proposes to construct the pseudo-demonstrations with the correct input distribution and label space (and format), and that paper described the "responses of LLMs are more likely to follow the labels of demonstrations that are semantically close to the input" as "copy effect".

It might be good to extend its analysis to include a deeper examination of the label distribution in in-context learning demonstrations, which potentially offer a more substantial contribution to the field.

---

> ### Author Rebuttal · Authors · 2024-05-31
>
> ### Response to Reviewer xiZm (1/2)
>
> Thank you for providing your valuable feedback. We have carefully considered your questions and would like to address them as below:
>
> > **w1： A series of recent works have specified that the formats and domains are important.**
>
> As we stated in Appendix E, while our work and previous works like Min et. al. 2022 and Pan et. al. 2023 all use the same term “format”, **our definition and research focus is different**. The experiments and analytical scope of Min et al. and Pan et al. can be interpreted as how format of demonstrations (query-answer pairing) affect performance, specifically: 1) the number of demonstrative query-answer pairs used, 2) whether the demonstrative query is paired with (correct) answer. In contrast, we specifically study the label space and format, and identify the regulation effect brought by ICL.
>
> We find that a significant portion of ICL's contributions stem from the label space and the format's regulatory effect on model responses, which is absent in previous works.
>
> > **w2: Some conclusions are already discussed in Z-ICL**
>
> Thank you for bringing to our attention the important work by Lyu et al. 2023. You are right that our paper would be enriched with an expanded discussion with Lyu et al., we will include the discussion and cite the paper in the final version. **Here we summarize the difference between our work and Lyu et al. as the following**:
>
> (1) Difference of target aspects being studied. The "Copy Effect" from Z-ICL is discussed under the context of using incorrect labels in demonstrations. As we mentioned in Section 2 (related works), all previous works focus on the label correctness, and take it as the start point to study the ICL. Instead, we decompose the benefit from ICL into three factors and take it as the start point to study which aspect ICL contributes to.
>
> (2) We take a deeper look into the "Copy Effect" from the perspective of majority label class instead of false labels. As stated in Section 6.2, we find that when providing demonstrations with the same class of current query's ground truth answer ("cheat" setting), we can observe ISIF percentage decreases, indicating the powers of label space and format are weakened when all demonstrations have the same label (especially semantically close to the input). This finding underscores diversity also plays critical role in selecting demonstrations.

---

> > ### Author Response · Authors · 2024-05-31
> >
> > ### Response to Reviewer xiZm (2/2)
> >
> > > **w3: It’s better to include a deeper examination of the label distribution in ICL.**
> >
> > Thank you for the suggestion. We would like to first emphasize that the main target and contribution of this work is to decompose the contribution of ICL into label space, label format and discrimination, and study each factor by designing comprehensive experiments. Whereas "label distribution" is not under our main scope, we tried our best to interpret the meaning behind "label distribution" pointed out by the reviewer and connected it with our study. We came up with the following interpretations.
> >
> >
> > (1) If "label distribution" refers to the distribution of different Ground Truth label (e.g., 30\% positive, 70\% negative), without including any false label, we feel that the analysis in Section 6 regarding the usage of "cheat" demonstrations could serve a part of impact of label distribution, as "cheat" settings destroy diversity of demonstration labels. The "cheat" experiments reveals that diversity of labels plays critical role in selecting demonstrations, evidenced by the observation that ISIF percentage decreases for both "cheat" settings.
> >
> >
> > (2) If "label distribution" refers to the rate of randomly assigned non-ground-truth labels (e.g., 70\% Ground-Truth, 30\% False Label), we think this problem is beyond our research scope, since the topic has been well studied, e.g. Yoo et. al. 2022 have conducted detailed analysis on different levels of false labels and measured the sensitivity of label correctness in demonstrations.
> > In section 5.3, we conduct another set of experiments by replacing all the ground-truth labels within demonstrations to incorrect labels.
> >
> > By decomposing the label space and format, we find that when provided with incorrect labels within the demonstrations, the ICL’s power of regulating label space and format is barely influenced. This observation can explain the reason that incorrect labels within demonstrations have minimal impact on overall performance.
> >
> > Thank you for your insightful feedback again!

---

> ### Author Response · Authors · 2024-06-05
> **More Discussion with Reviewer xiZm**
>
> Dear Reviewer xiZm,
>
> Thank you for your valuable feedback on our work. We have thoroughly discussed your concerns regarding Z-ICL (by Lyu et al.) and the "copy effect."
>
> As the discussion deadline approaches, **we are keen to know if our rebuttal has addressed your concerns and we welcome any further comments and suggestions.**
>
> Thank you again for your insightful contributions.
>
> Best regards,
>
> Authors

---

### Official Review · Reviewer_tah4 · 2024-05-13

**Rating:** 8
**Confidence:** 4
**Ethics Flag:** 1

**Summary:**

This paper measures the power of ICL by distinguishing its improvement on making models responding to the possible classification labels (label space), responding with exactly the right string (label format), vs. model's ability to classify correctly (discrimination). The paper finds that much of ICL contributes to improvement in label space and format, not discrimination. The paper also finds that writing detailed instructions, even zero-shot without demonstrations, can be just as good as ICL or ICL + detailed instructions.

**Questions To Authors:**

1. I didn't realize that experiments before 5.2 do not contain detailed instructions. Maybe this is a little unfair that models have to second guess the label format, maybe not. But at least the non-detailed-ness should be explicitly mentioned earlier.
2. Looks like DI, ICL, and DI-ICL has no major differences except for Llama 2. Can there be a dataset breakdown for figure 4? (I guess that's what Appendix F is for, but a figure in the main text would be even nicer. Thanks!)
3. Just a writing suggestion, take it or leave it: it might be even simpler to explain your definitions of Label Space := all acceptable labels regardless of synonyms. Label Format := exact string match. If using a perfect model parser/human rater to eval all examples, then there wouldn't be any issue with label formatting.

**Reasons To Accept:**

The paper is clearly written and I find Section 3.2's proposed distinctions on label space, label format, and discrimination as well as Section 3.1's distinctions on in-space, out-of-space, in-format, out-of-format to be helpful and productive, especially with the NLI examples. The experiments are clean and the analyses are clear, with multiple datasets with multiple models. The ablations in Section 6 are helpful and interesting too. The takeaways will also be useful to a lot of ICL users.

**Reasons To Reject:**

None specifically.

---

> ### Author Rebuttal · Authors · 2024-05-31
>
> Thanks for your valuable comments, helpful suggestions and appreciation of our work!
>
> > **Q1: Experiments before 5.2 should mention that prompts do not contain the detailed instructions. Unfairness that the model need to guess the label format.**
>
> Thanks for your suggestion! We will modify our manuscript and give more details about the prompts in experiments before Section 5.2. In regard of the unfairness that the model need to guess the label space and format, we follow several previous works [Wang et. al. 2023](https://aclanthology.org/2023.emnlp-main.609/), [Min et. al. 2022](https://aclanthology.org/2022.emnlp-main.759/), [Pan et. al. 2023](https://aclanthology.org/2023.findings-acl.527/), [Yoo et. al. 2022](https://aclanthology.org/2022.emnlp-main.155/)) that do not contain the detailed instructions including the label space and format in their prompt template.
>
> While it is indeed challenging to guess the space and format without detailed instructions, this setup is crucial to make sure we can decompose the contribution of ICL into different dimensions including label space, format and discrimination which has not been studied in previous works.
> Therefore, it’s necessary to conduct experiments with decomposed metrics (section 5.1) and with detailed instructions (section 5.2). By comparing the results in section 5.2, we can conclude that ICL functions similar to detailed instructions.
>
> > **Q2: Detailed breakdown of dataset in the main text would be nicer**
>
> Thank you for your suggestion! As the results in the main text are averaged scores, detailed scores for each model and dataset are provided in Appendix F. We will include some key tables in the main text in the revised version. On average, there is no big difference among DI, ICL and DI-ICL, confirming our claim that ICL acts similarly to detailed instructions in regulating the label space and format.
>
> > **Q3: Simpler to explain the concept of OOF and OOS. Label Space := all acceptable labels regardless of synonyms. Label Format := exact string match.**
>
> We appreciate your concise suggestion on the definition of Label Format and Label Space, and we will definitely take your suggestion into account and revise our descriptions accordingly in the final version.
>
> Thank you for your insightful feedback again!

---

### Decision · Program_Chairs · 2024-07-10

**Decision:**

Accept

**Comment:**

Paper is interesting and reviewers liked it. There are also champions for this paper.

I skimmed the paper myself and thought this is good work with good thinking behind. The framing and effort being also looks very scientific. It's also interesting to really break down and try to understand ICL.

I recommend acceptance.